# Revisiting Multilingual Data Mixtures in Language Model Pretraining

## Abstract

The impact of different multilingual data mixtures in pretraining large language models (LLMs) has been a topic of ongoing debate, often raising concerns about potential trade-offs between language coverage and model performance (*i.e.*, the curse of multilinguality). In this work, we investigate these assumptions by training 1B and 3B parameter LLMs on diverse multilingual corpora, varying the number of languages from 25 to 400. Our study challenges common beliefs surrounding multilingual training. First, we find that combining English and multilingual data does not necessarily degrade the in-language performance of either group, provided that languages have a sufficient number of tokens included in the pretraining corpus. Second, we observe that using English as a pivot language (*i.e.*, the language with the highest data proportion) yields benefits across language groups, and contrary to expectations, selecting a pivot language from within a specific group does not consistently improve performance for languages within that language branch. Lastly, we do not observe a significant "*curse of multilinguality*" as the number of training languages increases in models at this scale. Our findings suggest that multilingual data, when balanced appropriately, can enhance language model capabilities without compromising performance, even in low-resource settings.[1]

## 1 Introduction

Recent advances in large language models (LLMs) have demonstrated impressive performance across a wide range of non-English languages, including many that are considered low-resource (Yang et al., 2025; Team et al., 2025; Grattafiori et al., 2024; Üstün et al., 2024; OpenAI et al., 2024). These models are typically pretrained on data from over 100 high- and mid-resource languages, leveraging the broad availability of multilingual content on the web. Despite this progress, the impact of multilingual data composition on model training remains a subject of active debate, particularly regarding potential trade-offs between total language coverage and model performance in different languages (Alastruey et al., 2025). Practitioners often face difficult trade-offs: Should they include more languages in the pretraining data mixture or concentrate resources to prioritize performance in fewer languages? For greater multilingual generalization, should they include pivot languages from different language families or merely from high-resource global languages? Could curriculum learning among pivot languages also lead to greater multilingual generalization?

While previous studies tried to address these questions, they have generally been limited in scope, either by the number of languages considered or by the scale of the models used. For instance, one study investigates the so-called *curse of multilinguality* using relatively small models with 45M parameters (Chang et al., 2024). Another recent work explores scaling laws for multilingual language models and proposes an optimal sampling ratio for multilingual data (He et al., 2024). However, this work focuses on only 23 languages and similarly small models (85M parameters). Other studies have discussed multilingual data mixtures for task training (Wang et al., 2020) or instruction-tuning (Üstün et al., 2024), but it is unknown to what extent their intuitions would extend to pretraining.

In this work, we study the impact of multilingual data composition in training large-scale LLMs. Specifically, we train a series of 1B and 3B parameter models on corpora of 100B tokens containing up to 400 languages, allowing us to systematically explore the effects of language count, diversity,

---

[1]We will make our code available upon publication.

and token distribution. Our experiments challenge several prevailing propositions about multilingual training. We summarize our key findings as follows:

**Findings #1: More English data does not necessarily hurt multilingual performance.** We show that varying the proportion and absolute amount of English data in the training mix does not harm multilingual performance, as long as a sufficient number of multilingual tokens are included in the pretraining mixture. The reverse is also true, as increasing the number of multilingual tokens does not harm English performance as long as there are sufficient English tokens in the pretraining mixture.

**Findings #2: Typological boundaries are not barriers to transfer.** Contrary to the prevailing wisdom that family-specific pivots are most effective (He et al., 2024; Bagheri Nezhad & Agrawal, 2024), we find that using English as a pivot language[2] provides benefits across distinct linguistic groups. Selecting a high-resource pivot language from within a specific language branch (*e.g.*, Russian for Slavic languages) does not consistently enhance performance across languages in that branch. Given that English has the most diverse and highest quality data on the web, this evidence shows the unique advantage of leveraging a high-resource language to improve performance in other languages, regardless of their specific linguistic branch or structural typology.

**Findings #3: Curriculum learning fails to mitigate negative interference.** Prior work has shown training on multiple languages simultaneously can degrade performance in both high- and low-resource languages, a phenomenon coined as negative interference (Wang et al., 2020). Although curriculum learning has been proposed as a potential solution to this problem (Zhang et al., 2021; Kumar et al., 2021; Choi et al., 2023), our results show that staging the introduction of languages during training neither reduces negative interference nor improves performance on non-English languages.

**Findings #4: Increasing the number of training languages does not always lead to performance degradation.** The *curse of multilinguality* suggests that expanding language coverage reduces model performance in both monolingual and cross-lingual settings (Chang et al., 2024; Blevins et al., 2024; Pfeiffer et al., 2022; Conneau et al., 2020). We find the *curse of multilinguality* arises not from simply adding more languages, but from the finite capacity of models and data distributions that amplify the impact of noisy, low-resource languages.

Collectively, our findings offer practical guidance for designing more effective multilingual pretraining strategies and contribute to the development of stronger, more inclusive multilingual LLMs.

## 2 EXPERIMENTAL SETUP

**Model.** We train decoder-only Transformer models (Vaswani, 2017) based on the LLaMA architecture (Touvron et al., 2023), in two sizes: 1.1 and 3 billion parameters (1.1B and 3B). The model sizes are determined by varying the number of layers, hidden dimensions, and attention heads. Detailed configuration and training parameters are provided in Appendix A.

**Pretraining Data.** We use two corpora in our experiments. For experiments involving 30 languages, we use the multilingual version of the C4 corpus (mC4; Xue et al., 2021; Raffel et al., 2019).[3] For experiments involving a larger set of up to 1,834 languages, we use the FineWeb2 corpus (Penedo et al., 2025). All data are tokenized using the `Mistral-Nemo-Base-2407` tokenizer,[4] which has a vocabulary size of $|\mathcal{V}| = 131{,}000$ tokens. Models are trained on $D = 100$ to $D = 225$ billion tokens. We selected the `Mistral-Nemo-Base-2407` tokenizer because it is a state-of-the-art tokenizer designed specifically for multilingual pretraining, covering a wide range of scripts and languages (over 100), and representing them more fairly than other publicly available tokenizers (Apertus Project et al., 2025).

**Evaluation.** We evaluate our models by measuring their language modeling loss on a held-out validation set that is distinct from the pretraining data. In addition, we perform *downstream task evaluations* using a suite of multilingual benchmarks. For each model, we aggregate results by

---

[2]Historically, *pivot* languages are used as intermediary languages for *many-to-many* translation. In the context of this work we refer to pivot languages as those that are highly represented in pretraining data and whose presence serves as a catalyst for multilingual generalization.

[3]https://huggingface.co/datasets/allenai/c4

[4]https://huggingface.co/mistralai/Mistral-Nemo-Base-2407

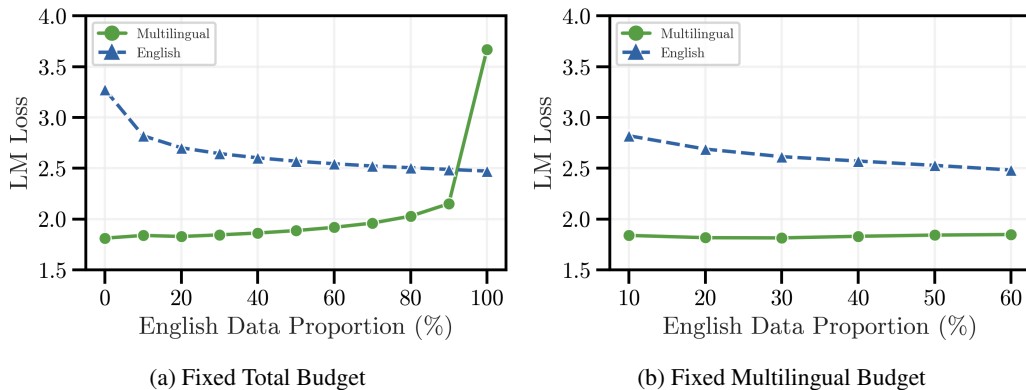

Figure 1: Validation *LM loss* for **English** and weighted average *LM loss* of non-English languages (**Multilingual**) across different proportions of English in the pretraining data for **1.1B** models. **(a)** In a **Fixed Total Budget**, increasing English data ($\geq$50%) leads to a performance drop in other languages. **(b)** In a **Fixed Multilingual Budget**, increasing English data (up to 60%) does not have a negative effect on other languages.

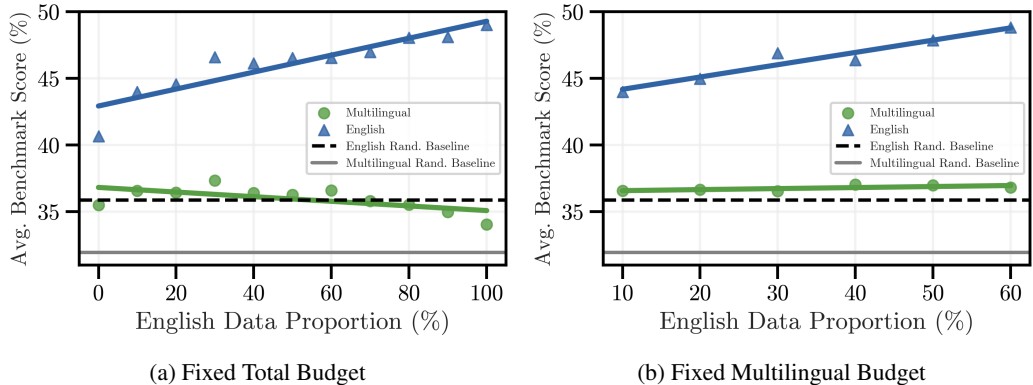

Figure 2: Aggregated *benchmark performance* for **English** and weighted average of non-English (**Multilingual**) across different proportions of English in the training data for **1.1B** models. The dashed lines represent the random baselines for each language group. **(a)** In a **Fixed Total Budget**, increasing English data ($\geq$50%), does not hurt downstream performance on the **Multilingual** group. **(b)** In a **Fixed Multilingual Budget**, we see that increasing English data has a negligible impact on the **Multilingual** group's performance.

language to obtain a comprehensive score for every model-language pair. Details of the benchmark suite and the aggregation procedure are provided in Appendix B.

## 3 ASSUMPTION #1: ENGLISH HURTS MULTILINGUALITY

English serves as the dominant pivot language for LLMs due to the abundance, diversity, and quality of English data available on the web. Simultaneously, due to the prevalence of LLM applications in English, maintaining English performance is often prioritized when training multilingual models by increasing the total proportion of English data, potentially at the expense of multilingual performance (Chung et al., 2023; Xue et al., 2021; 2022).

> **Assumption 1:** More English data comes at the cost of performance in other languages.

In this experiment, we investigate how the amount of English pretraining data influences performance in non-English languages. We train models of 1.1B and 3B parameters using data in 30 languages

from the mC4 corpus, systematically varying the proportion of English data from 0% to 100%. The selected languages represent diverse language families and data resource levels (Table 3). We use temperature sampling with $\tau = 3.3$ (details in Appendix A.3). When deciding on the data budget for these experiments, we consider two settings to disentangle the impact of data composition from the total amount of data seen during training:

*Fixed Total Budget*: The total pretraining budget is held constant at 100B tokens. Increasing the proportion of English reduces the amount of non-English (multilingual) data. This setup explores the trade-off between English and multilingual data under a constrained data regime.

*Fixed Multilingual Budget*: The amount of non-English data is fixed at 90B tokens with English data added on top, leading to a growing total data size (up to 225B tokens). This setup explores the effect of increasing English data without reducing multilingual coverage, simulating an unconstrained data regime (where multilingual data may be available in smaller quantities in web data than English data).

**Results.**     Figure 1a shows the final validation loss for English and non-English languages for the **1.1B** model for the *Fixed Total Budget* setting. As expected, increasing the proportion of English data leads to a lower validation loss for English. For non-English languages, validation loss remains relatively stable up to approximately 40% English data. Beyond this point, performance begins to degrade, indicating that allocating more capacity to English at the expense of other languages negatively impacts multilingual learning.

In contrast, under the *Fixed Multilingual Budget* setting (Figure 1b), we observe that multilingual performance remains largely unaffected—even when English comprises up to 60% of the dataset. These results suggest that, provided there is sufficient data to support learning robust multilingual representations, adding more English data does not interfere with performance on other languages. A similar pattern holds for the 3B models, as shown by the results in Appendix Figure 7.

Figure 2b presents the benchmark results for this experiment. In both the *Fixed Total Budget* and *Fixed Multilingual Budget* settings, we observe that increasing the proportion of English data consistently improves downstream task a in English. Mirroring the same patterns as for the loss, this increase does not degrade performance on other languages. Furthermore, Figure 8 shows results for the 3B models (see Appendix C), which exhibit a similar trend.

**Takeaway:** Contrary to common belief, increasing the amount of English data in the training of LLMs does not necessarily degrade their multilingual capabilities, provided that the training set also contains a sufficient quantity of multilingual tokens. In other words, it is possible to support additional languages while still maintaining strong performance in English.

## 4    ASSUMPTION 2: "STAY IN THE LANGUAGE BRANCH"

Previous research suggests that cross-lingual transfer is generally more effective between languages that belong to the same language family (Muller et al., 2023; He et al., 2024; Bagheri Nezhad & Agrawal, 2024; Xu et al., 2025). This implies that, if the pattern holds consistently, selecting a pivot language from within the same family is likely to yield greater transfer benefits than choosing one from a different family.

> **Assumption 2:**  Languages within the same linguistic branch offer the strongest boost to multilingual generalization.

In this experiment, we investigate the impact of using various types of pivot languages in a training corpus with multiple language families. A pivot language is defined as an intermediary language in a pretraining set for more effectively learning languages with less available data.

We compare using English as a pivot language for all languages, and selecting a pivot language from within the same language family for certain languages. Specifically, we train a 1.1B model on a subset of *Slavic* and *Cyrillic-script* languages under three different conditions: (1) English as the pivot language, (2) Russian as the pivot language, and (3) a uniform combination of English and Russian as pivots. The *Slavic* set includes Belarusian, Ukrainian, Macedonian, Bulgarian, Mongolian, Serbian,

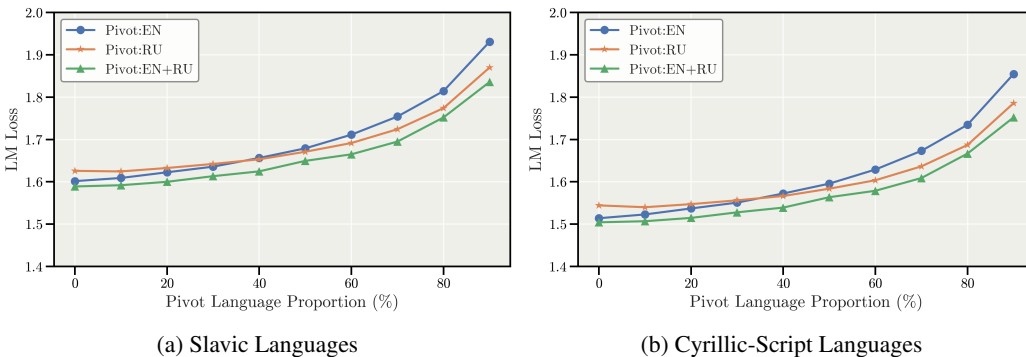

(a) Slavic Languages  (b) Cyrillic-Script Languages

Figure 3: Weighted average of validation LM loss for (a) *Slavic* and (b) *Cyrillic-script* languages when we have English, Russian, or English+Russian as a pivot language in the training data mix. Having a combination of Russian and English as pivots leads to the best performance for both groups of languages (Model size = 1.1B).

Polish, Czech, and Slovak. The *Cyrillic-script* set comprises Belarusian, Ukrainian, Macedonian, Bulgarian, Kyrgyz, Tajik, Kazakh, Mongolian, Serbian, and Uzbek (see Table 4 for details).

**Results.** Figure 3 presents the weighted average loss across both language groups. We observe that as the proportion of training data assigned to the pivot language increases (and the complement proportion for non-pivot languages decreases), the loss for non-pivot languages remains relatively stable at first. However, as less data is allocated to them, their loss eventually rises, as expected. Up to a 50% allocation to the pivot language, English and Russian perform comparably. However, beyond this threshold—particularly at 60% or more, Russian proves slightly more effective as a pivot, yielding lower loss for the remaining languages. One possible explanation is that when pivot allocation is relatively low, non-pivot languages still benefit from having access to their own training data. But in extremely low-resource conditions, these languages gain more from leveraging similarities with a strong pivot language. Another factor is that English training data is often more diverse and standardized, with broad domain coverage. This richness may make English a strong pivot up to a certain point, after which typological proximity favors Russian. Notably, combining *both* English and Russian as joint pivots yields the lowest overall loss, suggesting a complementary effect: English contributes wide coverage, while Russian offers closer linguistic ties to many of the target languages. The detailed per-language loss values are provided in Figure 9 in Appendix C.

**Takeaway:** English can serve as a broadly effective pivot language, but in very low-resource settings, typological similarity becomes increasingly important. Using multiple pivots that balance breadth and proximity provides the most consistent benefits across language families.

## 5 ASSUMPTION 3: MULTILINGUAL CURRICULUM LEARNING REDUCES NEGATIVE INTERFERENCE

Prior research has explored using curriculum learning (a "general-to-specific" data scheduling approach) to improve pretraining (Dubey et al., 2024; DataBricks, 2024; Apertus Project et al., 2025; Martins et al., 2025). In multilingual training, previous work suggests that the order in which languages are introduced during training can influence model performance and potentially reduce competition between languages (Choi et al., 2023; Ranaldi et al., 2024; Allemann et al., 2025).

> **Assumption 3:** Curriculum-based language introduction mitigates negative interference.

To investigate the dynamics of cross-lingual competition and knowledge transfer in multilingual language models, we designed a series of controlled *curriculum learning* experiments. Our goal is to understand how the timing and order of language inclusion during training influence model performance. We design four experimental setups:

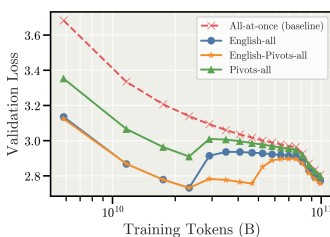 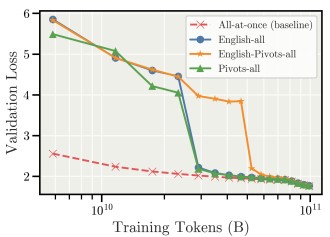 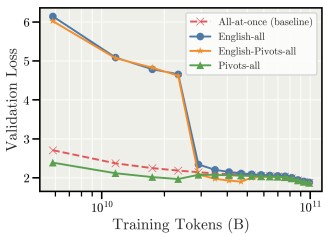

(a) Loss on English-language validation data.

(b) Weighted average loss for non-pivot languages.

(c) Weighted average Loss for Arabic, Chinese, and Russian.

Figure 4: LM loss on the validation set for $3B$ models as a function of consumed training tokens, shown separately for (a) English, (b) non-English, and (c) pivot languages under different curriculum strategies.

*All-at-once baseline:* The model is trained on the full multilingual dataset from the outset. This setup, common in many multilingual LLMs (*e.g.*, Apertus Project et al., 2025) serves as a control to benchmark the effects of curriculum-based training strategies.

*English-all*: For the first 25% of training, the model is exposed only to English. After this phase, training proceeds on the full multilingual dataset. This allows us to isolate the impact of early single-language pretraining on subsequent multilingual generalization and interference.

*English-Pivots-all*: Training is divided into three phases (1) 0–25%: Only English data is used. (2) 25–50%: We introduce three additional high—resource languages—Arabic, Chinese, and Russian—alongside English as pivot languages. These four languages were chosen to represent four distinct scripts: Latin, Arabic, Han, and Cyrillic, respectively. This intermediate stage allows us to explore early competition between strong languages with differing orthographic and typological properties. (3) 50–100%: The model is trained on the full multilingual dataset. This progressive inclusion strategy enables a controlled examination of cross-lingual interactions and competition under varying degrees of language diversity.

*Pivots-all*: For the first 25% of training, the model is trained using our 4 pivot languages. After this phase, training continues on the full multilingual dataset. This allows us to isolate the impact of early high-resource pretraining on subsequent multilingual generalization and interference.

**Results.**   Figure 4 presents the results of our curriculum learning experiments. When examining English loss, we find that introducing English early in training—either alone (*English-all*) or alongside pivot languages (*English-pivots-all*)—leads to lower final loss for English. Notably, transitioning between curriculum stages (*i.e.*, adding new languages in successive phases) temporarily increases the loss for previously seen languages. This suggests a short-term "forgetting" effect, where the model learns new languages at the cost of temporarily degrading performance on earlier ones, before eventually recovering and integrating all knowledge across the languages.

For the other three pivot languages (Figure 4c), the curriculum that begins with English and subsequently introduces the pivots (*Pivots-all*) achieves the lowest average loss midway through training. However, as additional languages are introduced, the loss increases, ultimately converging to the same level as other runs. As with English, we observe a forgetting effect at each transition.

When analyzing the average loss across other non-English languages (Figure 4b), we observe that while different curriculum regimes begin at different starting points and follow distinct learning trajectories, they all converge to a similar final loss by the end of training. This consistency indicates that curriculum order primarily affects learning dynamics, but not final multilingual performance.

Although curriculum learning appears to benefit English, further analysis reveals that this improvement is largely attributable to data quantity. Specifically, we find a strong correlation between the number of English tokens in the training mix and the model's performance on English. In other words, models exposed to more English data achieve lower loss. Consequently, the *English-pivots-all* setup attains the lowest English loss primarily because it includes the largest proportion of English data in its curriculum. Validation loss for each language is depicted in Figures 11 and 12 in Appendix E.

**Takeaway**: Curriculum learning shapes the trajectory of multilingual training but does not reduce interference or improve final performance. Observed gains for English under certain curricula are explained by the data distribution rather than curriculum structure.

# 6 ASSUMPTION 4: THE "CURSE OF MULTILIGUALITY"

Prior work has shown that, for a fixed model capacity, adding more languages during pretraining initially improves cross-lingual transfer, particularly for low-resource languages. However, beyond a certain point, both monolingual and cross-lingual performance begin to degrade. This trade-off is commonly referred to as the *curse of multilinguality* (Conneau et al., 2020; Pfeiffer et al., 2022; Blevins et al., 2024; Chang et al., 2024).

> **Assumption 4:** Adding more languages to a pretraining mixture reduces performance.

We revisit this assumption by training language models with varying numbers of languages and analyzing the impact on both high- and low-resource languages.

| # Languages | LM Loss ↓ | | Benchmark Performance ↑ | |
|:---:|:---:|:---:|:---:|:---:|
| | **Natural Dist.** | **Temp. Sampling** | **Natural Dist.** | **Temp. Sampling** |
| 25 | 2.678 | 2.675 | $50.13 \pm 1.868$ | $43.24 \pm 1.874$ |
| 50 | 2.678 | 2.681 | $49.41 \pm 1.868$ | $43.80 \pm 1.878$ |
| 100 | 2.682 | 2.687 | $49.29 \pm 1.865$ | $43.76 \pm 1.872$ |
| 200 | 2.680 | 2.696 | $49.11 \pm 1.864$ | $42.38 \pm 1.870$ |
| 400 | 2.678 | 2.707 | $49.64 \pm 1.871$ | $42.12 \pm 1.854$ |

Table 1: English validation loss and benchmark performance (%) when increasing languages coverage from 25 to 400 (3B model). English represents 40% of the training data in all runs (40B tokens). Increasing the number of languages, while keeping English data fixed, does not hurt English performance. Per-benchmark values are provided in Table 9.

Practically, we train 3B parameter models on 100B tokens from the FineWeb-2 corpus. In all settings, English accounts for 40% of the training data, while the number of non-English languages is systematically increased—from 25 to 400. We experiment with the top-25, 50, 100, 200, and 400 most frequent languages in FineWeb-2 under two distributions: (1) the *natural distribution* and (2) *temperature sampling* with $\tau = 3.3$. We then evaluate how increasing linguistic diversity in the non-English data subset affects English and non-English performance. Details of the training data distribution are provided in Table 8.

**Results.** Table 1 summarizes English validation loss and average downstream performance across these configurations. Two main observations emerge. First, for a fixed number of languages and a fixed English share, English performance is consistently stronger under the natural distribution than under temperature sampling. In this case, English benefits from cross-lingual transfer with high-resource, typologically related languages (*e.g.*, German, French), which receive more data under the natural distribution (we further investigate this effect in Appendix D). Second, even when scaling up to 400 languages, English performance remains largely stable—particularly under the natural distribution, suggesting that English performance is not determined by the sheer number of languages included in the training process. In other words, the key factor is not how many languages are present, but how the training data is distributed among them.

Building on this insight, we show in Figures 5a and 5c the weighted average LM validation loss for the top-25, 50, 100, and 200 language groups (excluding English) under the two distributions. The $x$-axis denotes language groups used for evaluation, while the $y$-axis indicates language groups used for training. Because the total data budget is fixed at 100B tokens, adding more languages necessarily reduces the relative share of data for previously included ones. Under the natural distribution, however, performance remains stable as languages are added. In contrast, under temperature sampling we observe up to a $\sim$0.1 increase in loss when expanding from 25 to 400 languages. This effect is

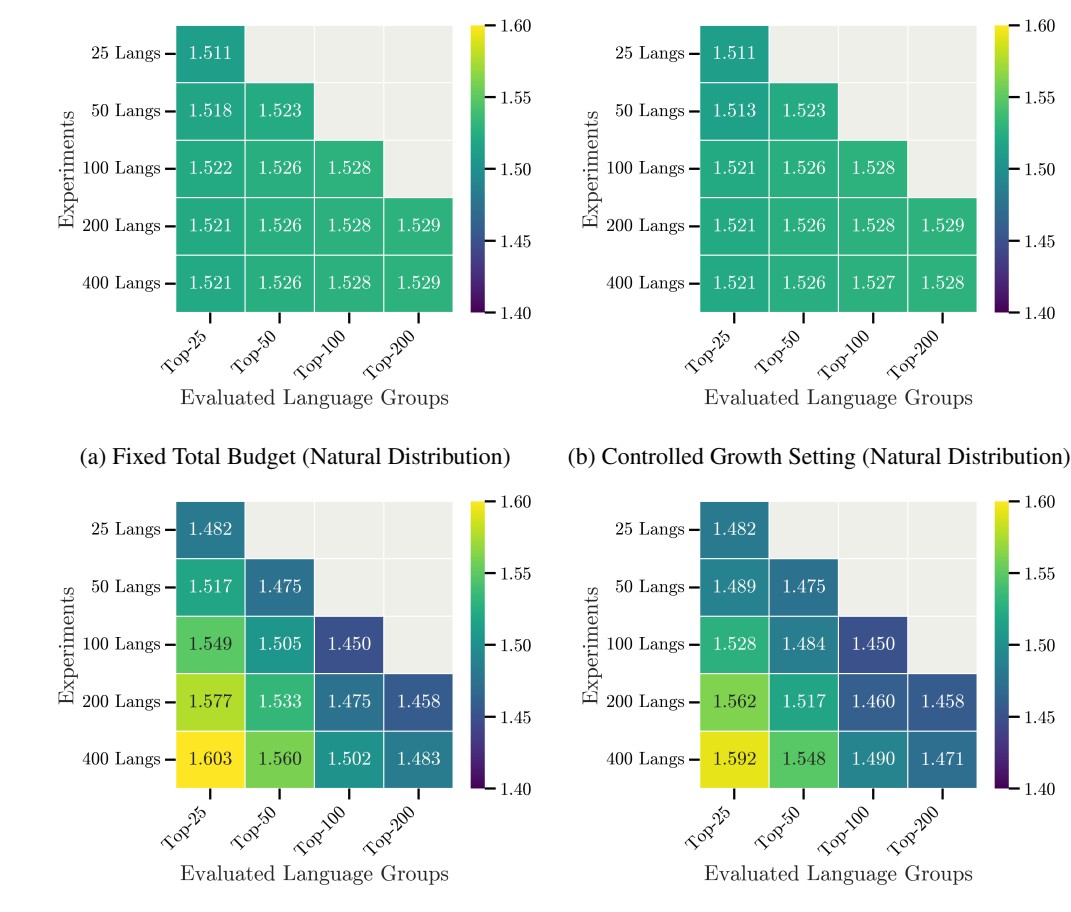

(a) Fixed Total Budget (Natural Distribution)     (b) Controlled Growth Setting (Natural Distribution)

(c) Fixed Total Budget (Temperature Sampling)     (d) Controlled Growth Setting (Temperature Sampling)

Figure 5: Average validation LM loss for different language groups ($x$-axis) across various *curse of multilinguality* experiments that include more languages in the pretraining mixture ($y$-axis). Increasing the number of languages does not necessarily degrade the performance of languages included in previous experiments, provided that the amount of training data (in tokens) for those languages remains the same. (English is excluded from these evaluations)

expected, since temperature sampling reduces the allocation of mid- and high-resource languages more aggressively, amplifying the effect of including low-resource ones.

To disentangle the effect of adding new languages from the effect of reducing data for existing ones, we also run a controlled setting where the data for the original set of languages remains fixed across two consecutive runs. For example, when increasing from 25 to 50 languages, the first 25 languages receive the same amount of training data as before; the same approach is applied when scaling from 50 to 100, 100 to 200, and 200 to 400 languages. Figures 5b and 5d report the results. Once again, under the natural distribution, performance remains stable, and we also observe a smaller relative degradation for the temperature sampling setting.

Taken together, these results suggest that the *curse of multilinguality* is not primarily about the number of languages added, but instead reflects limitations in model capacity and the quality and distribution of multilingual data. Under the natural distribution, the phenomenon is better described as a *curse of capacity*: models have a finite ability to absorb tokens, and beyond a certain point, additional data yields diminishing or even negative returns, a constraint not unique to multilingual models. Under temperature sampling, the issue more closely resembles a *curse of data quality*: oversampling very low-resource languages introduces more noisy data into training, which hurts performance.

**Takeaway:** The *curse of multilinguality*, while measurable, likely arises not from simply adding more languages, but from (1) the finite capacity of models and (2) data distributions that too strongly amplify the impact of languages represented by lower-quality data.

## 7 RELATED WORK

**Pretraining Data Mixture.** Prior work has explored the impact of pretraining data composition on the performance of large language models (LLMs) (Gu et al., 2024; Zhao et al., 2024b; Xie et al., 2023; Albalak et al., 2023; Held et al., 2025; Apertus Project et al., 2025). Several studies have proposed algorithms to optimize domain weights using proxy models, thereby improving the generalization ability of LLMs (Xie et al., 2024; Fan et al., 2023). Another approach formulates the identification of high-performing data mixtures as a regression problem (Liu et al., 2024).

In the multilingual setting, temperature-based sampling has traditionally been used to balance representation across languages (Devlin et al., 2019; Xue et al., 2021). However, this heuristic method can lead to overfitting on low-resource (tail) languages. To address this, Chung et al. (2023) proposes a sampling method that ensures more uniform coverage of high-resource (head) languages while capping repetition on low-resource languages. Additionally, He et al. (2024) investigates scaling laws specific to multilingual LLMs, providing further insight into optimal data mixture strategies. Additionally, several approaches address data mixture optimization within the context of multilingual continual pretraining (Ji et al., 2024; Li et al., 2025).

**Curse of Multilinguality & Negative Interference.** The curse of multilinguality, introduced by Conneau et al. (2020), describes the phenomenon where, under a fixed model capacity, adding more languages initially improves cross-lingual performance—especially for low-resource languages—but eventually leads to degradation in both monolingual and cross-lingual performance. Most previous investigations into this phenomenon have been limited in scale, in terms of both model size and language coverage. For instance, Pfeiffer et al. (2022) studies this trade-off using a 270M parameter bidirectional model trained on 75 languages, proposing a modular architecture to mitigate interference. Recently, Chuang et al. (2025) shows that the curse of multilinguality breaks with larger count of parameters for multimodal embedding tasks.

Blevins et al. (2024) introduces a cross-lingual expert language model, in which separate models are trained on subsets of the multilingual corpus to reduce competition among languages. Similarly, Chang et al. (2024) explores this effect using monolingual and multilingual models (up to 45M parameters) trained across 250 languages and derives optimal sampling ratios. Wang et al. (2020) examines the phenomenon of negative interference in multilingual LMs and introduces a meta-learning algorithm that improves cross-lingual transfer and alleviates interference effects. Alastruey et al. (2025) challenge the prevailing assumption that cross-lingual interference depends on language family, showing instead that it is primarily related to script.

**Impact of Pivot Languages.** The role of *pivot* languages in improving monolingual and cross-lingual performance of multilingual LLMs has been studied before. Several works have demonstrated the benefits of using a pivot language for machine translation (Kim et al., 2019; Zou et al., 2022; Gaikwad et al., 2024; Mohammadshahi et al., 2024). Zhang et al. (2024) shows that using English as a pivot for cross-lingual instruction tuning, by first interpreting instructions in English before generating responses in the target language, can be highly effective. Pivot languages have also been used to improve alignment in multilingual representation spaces (Zhao et al., 2024a). Investigating the mechanics of this transfer, Wendler et al. (2024) show that models leverage the pivot language's internal circuits to process other languages. The efficacy of this cross-lingual transfer is tied to data distribution: it is more pronounced when models are trained using imbalanced language mixtures, rather than in a balanced setting (Schäfer et al., 2024).

**Curriculum Learning (CL) for LLMs.** Curriculum Learning (CL), a data-centric training strategy inspired by human learning processes, has been studied for improving the performance of LLMs (Naïr et al., 2024; Kim & Lee, 2024; Li et al., 2021). Several studies have demonstrated the effectiveness of CL in multilingual machine translation (Zhang et al., 2021; Kumar et al., 2021; Zhou et al., 2021; Choi et al., 2023). Ranaldi et al. (2024) applies the CL paradigm during the instruction-tuning phase of multilingual LLMs and reports notable improvements. Additionally, Yoo et al. (2024) proposes a code-switching-based CL strategy to enhance cross-lingual transfer capabilities in LLMs.

## 8 DISCUSSION & CONCLUSION

Our study investigates the influence of data mixture composition on multilingual large language model (LLM) pretraining, leveraging 1.1B and 3B models across up to 400 languages. Our findings challenge prevailing assumptions about multilingual pretraining, offering direct guidance for multilingual data mixture design. First, we demonstrated that the quantity and proportion of high-resource English data do not inherently compromise multilingual performance, provided a sufficient number of non-English tokens are present. This finding suggests practitioners should prioritize ensuring an adequate absolute volume of diverse, high-quality multilingual content over strictly reducing the high-resource component. Second, and contrary to suggestions that family-specific pivots are most effective, we established that English consistently serves as a high-quality pivot language, providing cross-lingual transfer benefits across linguistic groups.

We also provided new insights into core challenges of multilingual scaling: negative interference and the "*curse of multilinguality*." Our results showed that staging the introduction of languages through curriculum learning does not mitigate negative interference (Wang et al., 2020) or improve non-English performance, suggesting that interference is a fundamental problem related to the fixed capacity of models, and not merely one that can be fixed through different data curricula. Furthermore, our findings refine the understanding of the "*curse of multilinguality*"(Chang et al., 2024), demonstrating that performance degradation arises not from the simple count of languages added, but from the finite capacity of models and data distributions that amplify the impact of noisy, low-resource languages. Although prior studies (Conneau et al., 2020) mentioned the capacity limitation in multilingual modeling, this work distinguishes itself by conducting a comprehensive and integrated analysis of the phenomenon across its various facets.

Collectively, these findings translate into the following practices for training multilingual LLMs. First, adopting regimes taking into account the order of languages such as curriculum learning offer no demonstrable benefit over a well-mixed approach. Second, given the resilience to high English proportions, focus resource investment on scaling and cleaning low-resource data rather than on costly data balancing operations. Third, do not limit language coverage arbitrarily, as the curse is primarily a function of quality and quantity of the multilingual data, not language count. Our evidence implies that future efforts to break the curse should focus on including adequate high-quality data for each language. While these principles were established on 1.1B and 3B parameter models, future work must validate these trade-offs on larger models models to explore how increased model capacity potentially alters the non-linear relationship between data composition, interference, and performance.

### LIMITATIONS

Despite employing larger models and more data than prior work, our study remains far below the scale of frontier models such as Meta AI (2025); Guo et al. (2025), as operating at that scale would have prevented us from running the number of experiments necessary to draw reasonable conclusions within our computational constraints. Furthermore, we were unable to explore the impact of post-training and the effects of various data sampling strategies for the same reason. Lastly, the choice of our tokenizer may limit performance on lower-resource languages. We selected a pre-existing tokenizer that supported the greatest number of languages in our study, as training a tokenizer to support 1,834 languages is practically infeasible without substantially increasing the model's vocabulary size and the associated GPU memory requirements.

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

# A  LANGUAGE MODEL TRAINING

Here we provide details about training the language models used in our experiments.

| Model | Arch. | Layers | Hidden | Attn. Heads | RoPE $\theta$ | Vocab |
|-------|-------|--------|--------|-------------|---------------|-------|
| 1.1B | LLaMA | 24 | 1536 | 16 | 500,000 | 131,000 |
| 3B | LLaMA | 28 | 2496 | 24 | 500,000 | 131,000 |

Table 2: Overview of the architectural configurations for different model sizes.

Our experiments focus on models with 1.1 and 3 billion parameters (1.1 and 3B). All models follow the LLaMA architecture Touvron et al. (2023). The model size is determined by adjusting the number of layers, hidden sizes, and the number of attention heads (Details in Table 2).

## A.1  TRAINING HYPERPARAMETERS

We train our models using HuggingFace's Nanotron trainer.[5] The key training hyperparameters are as follows:

- **Learning Rate.** We use a learning rate of $8 \times 10^{-4}$ with linear warmup over the first 4% of training. A "1-sqrt" decay schedule (Hägele et al., 2024) is applied during the final 20%, as shown in Figure 6.

- **Optimizer.** All experiments use AdamW with $\beta = (0.9, 0.95)$ (Loshchilov, 2017).

- **Weight Decay.** We set the weight decay parameter to $\lambda = 0.1$ for regularization.

- **Batch Size.** The micro-batch size is fixed at 5 across all runs.

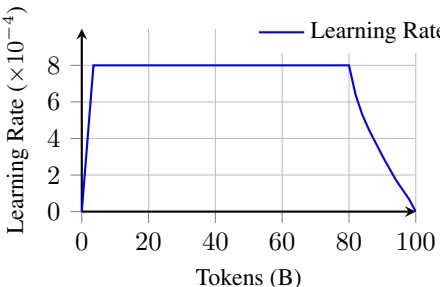

Figure 6: Learning rate schedule over tokens with warmup and decay.

---

[5]https://github.com/huggingface/nanotron

## A.2 HARDWARE SETUP

Training is performed on a large-scale cluster. Each node is equipped with 4 NVIDIA Grace-Hopper H100 GPUs (96 GB memory each).

- **1B models.** We train our 1B models on 22 nodes (or 88 GPUs) over around 15h per 100B tokens. This gives a global batch size of 440 examples.
- **3B models.** We train our 3B models on 64 nodes (or 256 GPUs) for around 18h per 100B tokens. Therefore our runs have a global batch size of 640 examples.

## A.3 SAMPLING METHODS

Let $\mathcal{L}$ be the set of languages in the dataset, and let $\pi^{\text{natural}} \in \Delta_{|\mathcal{L}|}$ represent the natural distribution of these languages, defined as:

$$\pi_l^{\text{natural}} = \frac{\omega_l}{\sum_{l' \in \mathcal{L}} \omega_{l'}}$$

where $\omega_l$ denotes the number of words (or tokens) for language $l$ in the dataset. In this work, we use the number of words as a proxy for language frequency, a common practice when presenting statistics for highly multilingual datasets Penedo et al. (2025). We implement the following sampling strategies:

- **Natural Sampling.** This method samples according to the natural distribution $\pi^{\text{natural}}$, directly reflecting language frequencies in the dataset. Typically, this distribution is highly imbalanced, with a few languages dominating the cumulative share of data.
- **Temperature Sampling.** This method adjusts the natural distribution using a temperature parameter $\tau$ to create a less skewed distribution:

$$\pi_l^{\text{temp},\tau} = \frac{\omega_l^{1/\tau}}{\sum_{l' \in \mathcal{L}} \omega_{l'}^{1/\tau}}$$

By tuning $\tau$, the distribution can be shifted towards uniformity, thereby reducing imbalance among languages.

Figures 10 and 11 present the training data distribution for experiments described in Section 3.

# B BENCHMARK SETUP

We evaluate our models using HuggingFace's Lighteval codebase (Habib et al., 2023).[6]

## B.1 BENCHMARKS

We select 10 standard multilingual benchmarks to evaluate our models on various downstream tasks.

- **Belebele**: A multilingual reading comprehension dataset containing passages and corresponding questions in many languages. It evaluates models' ability to understand text and answer related questions (Bandarkar et al., 2024).
- **XCodah**: A multilingual adaptation of CODAH for adversarially-authored commonsense reasoning tasks, testing robustness in natural language understanding (Lin et al., 2021a; Chen et al., 2019).
- **XCSQA**: A multilingual version of CommonsenseQA, consisting of multiple-choice questions that require reasoning about everyday concepts and their relations (Lin et al., 2021a; Talmor et al., 2019).
- **XCOPA**: A multilingual adaptation of the COPA dataset for evaluating cross-lingual causal commonsense reasoning, covering multiple languages to test reasoning transfer across linguistic boundaries (Ponti et al., 2020).

---

[6]https://huggingface.co/docs/lighteval/en/index

- **XStoryCloze**: A multilingual extension of the StoryCloze Test, where models must choose the most coherent ending to short narratives, testing story comprehension and commonsense reasoning (Mostafazadeh et al., 2017; Lin et al., 2021b).

- **XWinogrande**: A multilingual version of WinoGrande, containing sentences with ambiguous pronouns. It measures models' ability to resolve coreference using contextual and commonsense cues (Sakaguchi et al., 2021; Muennighoff et al., 2022; Tikhonov & Ryabinin, 2021).

- **MMMLU**: A multilingual adaptation of MMLU, evaluating model performance across a wide spectrum of tasks and domains (Hendrycks et al., 2021; Dac Lai et al., 2023).

- **INCLUDE**: A large-scale benchmark covering 44 languages, designed to evaluate multilingual LLMs in realistic language environments with a focus on knowledge and reasoning (Romanou et al., 2024).

- **Exams**: A benchmark of standardized test questions across subjects and educational levels, used to assess reasoning and problem-solving abilities in exam-like conditions (Hardalov et al., 2020).

- **M3Exams**: A multilingual exam-style benchmark that extends Exams across different languages, subjects, and difficulty levels (Zhang et al., 2023).

## B.2 AGGREGATIONS

We aggregate benchmark results to compute a language-specific score for each model. Let $\mathcal{T}_l$ be the set of benchmarks (or tasks) containing a split for language $l$. The aggregated score for a model $m$ per language $l$ is defined as:

$$s_l^m = \frac{1}{|\mathcal{T}_l|} \sum_{t \in \mathcal{T}_l} s_{t,l}^m$$

where $s_l^m$ is the score of a model $m$ on the split $l$ of a task $t$. To mitigate biases arising from varying numbers of benchmarks per language, we compute a language-specific random baseline $\zeta_l$. This baseline helps assess whether a given aggregated score significantly outperforms random predictions. Specifically, we calculate the random baseline for each language as the average of the individual random baselines across all tasks that include language $l$:

$$\zeta_l = \frac{1}{|\mathcal{T}_l|} \sum_{t \in \mathcal{T}_l} \zeta_t$$

## C  PIVOT ABLATION

Table 3 presents the languages included in the experiments described in Section 3. The set of languages analyzed in the experiments of Section 4 is listed in Table 4.

Figures 7 and 8 present the validation loss and average benchmark scores for English and non-English ("Multilingual") languages for **3B** models. Consistent with our observations for the 1.1B models, we find that under the Fixed Total Budget setting, increasing the proportion of English data ($\geq 50\%$), leads to a decline in performance for other languages. In contrast, under the Fixed Multilingual Budget setting, increasing the share of English data (up to 60%) does not adversely affect the performance of non-English languages.

## D  CROSS-LINGUAL TRANSFER

To examine how non-English languages influence English performance under the *Fixed Total Budget* setting, we train models on data spanning 1,834 languages while systematically varying the share of data allocated to each. Specifically, we partition the languages from the FineWeb-2 dataset into two groups:

***Target Languages.***  A set of 45 high- and mid-resource languages that we aim for the model to perform well on.

| Language | Language Family | Script |
|---|---|---|
| Arabic | Afro-Asiatic (Semitic) | Perso-Arabic |
| Bulgarian | Indo-European (Slavic) | Cyrillic |
| Bengali | Indo-European (Indo-Aryan) | Bengali |
| Catalan | Indo-European (Romance) | Latin |
| German | Indo-European (Germanic) | Latin |
| Greek | Indo-European (Hellenic) | Greek |
| English | Indo-European (Germanic) | Latin |
| Spanish | Indo-European (Romance) | Latin |
| Estonian | Uralic (Finnic) | Latin |
| Basque | Language Isolate | Latin |
| Persian (Farsi) | Indo-European (Iranian) | Perso-Arabic |
| Finnish | Uralic (Finnic) | Latin |
| French | Indo-European (Romance) | Latin |
| Hindi | Indo-European (Indo-Aryan) | Devanagari |
| Haitian Creole | Creole (French-based) | Latin |
| Indonesian | Austronesian | Latin |
| Italian | Indo-European (Romance) | Latin |
| Japanese | Japonic | Kanji & Kana (CJK) |
| Korean | Koreanic | Hangugeo (CJK) |
| Burmese | Sino-Tibetan | Burmese |
| Portuguese | Indo-European (Romance) | Latin |
| Russian | Indo-European (Slavic) | Cyrillic |
| Swahili | Niger-Congo (Bantu) | Latin |
| Tamil | Dravidian | Tamil |
| Telugu | Dravidian | Telugu (Brahmic) |
| Thai | Kra–Dai (Tai) | Thai |
| Turkish | Turkic | Latin |
| Urdu | Indo-European (Indo-Aryan) | Perso-Arabic |
| Vietnamese | Austroasiatic | Vietnamese (Latin-based) |
| Chinese (Mandarin) | Sino-Tibetan | Hanzi (CJK) |

Table 3: Languages used in experiments discussed in Section 3.

| Language | Language Family | Script |
|----------|-----------------|--------|
| English | Indo-European (Germanic) | Latin |
| Russian | Indo-European (Slavic) | Cyrillic |
| Ukrainian | Indo-European (Slavic) | Cyrillic |
| Belarusian | Indo-European (Slavic) | Cyrillic |
| Serbian | Indo-European (Slavic) | Cyrillic |
| Macedonian | Indo-European (Slavic) | Cyrillic |
| Bulgarian | Indo-European (Slavic) | Cyrillic |
| Polish | Indo-European (Slavic) | Latin |
| Czech | Indo-European (Slavic) | Latin |
| Slovak | Indo-European (Slavic) | Latin |
| Tajik | Indo-European (Iranian) | Cyrillic |
| Uzbek | Turkic | Cyrillic |
| Kyrgyz | Turkic | Cyrillic |
| Kazakh | Turkic | Cyrillic |
| Mongolian | Mongolic | Cyrillic |

Table 4: Languages used in experiments discussed in Section 4.

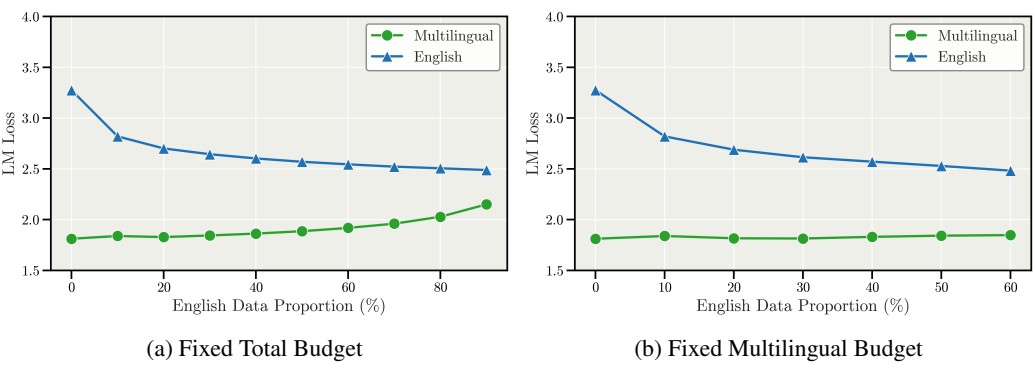

(a) Fixed Total Budget        (b) Fixed Multilingual Budget

Figure 7: Validation LM loss for **English** and weighted average LM loss of non-English (**Multilingual**) across different proportions of English in the training data for **3B** models. (a) In a **Fixed Total Budget**, increasing English data ($\geq$50%), leads to a performance drop in other languages. (b) In a **Fixed Multilingual Budget**, increasing English data (up to 60%) does not have a negative effect on other languages.

***Tail Languages.*** The remaining 1,789 low-resource languages, which the model is expected to support only as a secondary objective.

The full lists of target and tail languages are provided in Appendix D.1. Importantly, we exclude English from the training data to neutralize its dominant influence and allow for a clearer analysis of cross-linguistic interactions. We train 3B-parameter models by varying the proportion of tail-language data in the training mix, ranging from 6% to 33%, and evaluate the impact on performance across the target language set.

Figure 10a presents the effect of adjusting the balance between the top-25 high-resource languages (in FineWeb-2) and the remaining languages on English validation loss. Although English is not part of the training data, we observe that its validation loss decreases as more tokens from high-resource languages are included, and increases when more tokens from lower-resource languages are introduced. This effect is likely due to the close linguistic proximity of several high-resource languages (*e.g.*, German, French) to English, which provides beneficial transfer.

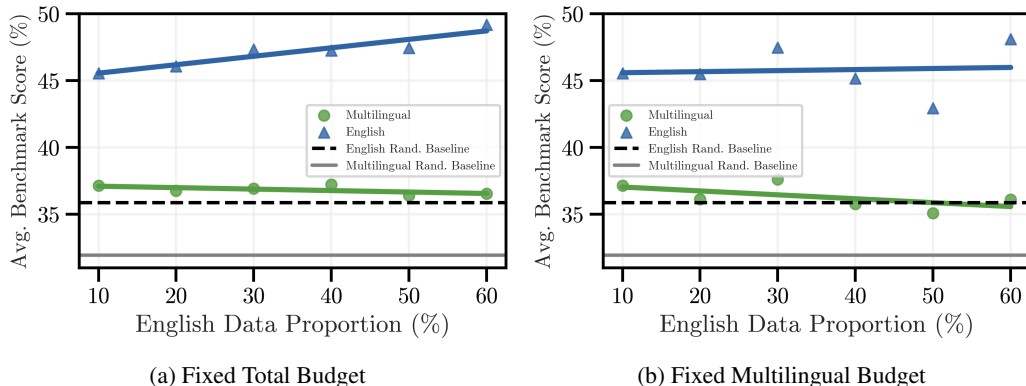

(a) Fixed Total Budget            (b) Fixed Multilingual Budget

Figure 8: Aggregated benchmark performance for **English** and weighted average of non-English (**Multilingual**) across different proportions of English in the training data for $3B$ models. The dashed lines represents the random baselines for each language group. **(a)** In a **Fixed Total Budget**, increasing English data ($\geq 50\%$), does not hurt downstream performance on *other* group. **(b)** In a **Fixed Multilingual Budget**, we see that increasing English data has a negligible impact on the *other* group's performance.

Supporting this interpretation, we find that English performance is most strongly correlated with Romance, Slavic, and Germanic languages, with Pearson correlation coefficients of 0.78, 0.85, and 0.80, respectively (Table 5). Figure 10b shows the same pattern in benchmark results: English benefits from the presence of related high-resource languages. Together, these findings highlight a positive interaction between English and typologically related high-resource languages, which enhances English performance even when it is excluded from training.

### D.1 TARGET AND TAIL LANGUAGES

The target languages used in the Curse of Multilinguality experiments are as follows: German, Russian, French, Japanese, Spanish, Mandarin Chinese, Italian, Dutch, Polish, Portuguese, Czech, Vietnamese, Indonesian, Turkish, Swedish, Persian (Farsi), Korean, Hungarian, Arabic, Greek, Romanian, Danish, Finnish, Thai, Ukrainian, Slovak, Norwegian Bokmål, Bulgarian, Catalan, Croatian, Latin, Serbian, Hindi, Slovenian, Lithuanian, Estonian, Hebrew, Latvian, Tosk Albanian, Icelandic, Macedonian, Galician, Basque, Malayalam, Romansh, Swiss German. Tail languages contain the rest of the languages from the FineWeb-2 corpus.

Tables 6 and 7 present detailed information about the language families and scripts included in the FineWeb-2 dataset.

## E CURRICULUM LEARNING

Figures 11 and 12 show the validation loss for each language for the experiments described in Section 5.

## F CURSE OF MULTILINGUALITY

Using a fixed total data budget, Table 15 reports the validation loss for 50 languages trained under natural distribution and temperature sampling conditions. The models used for this analysis are detailed in Section 6.

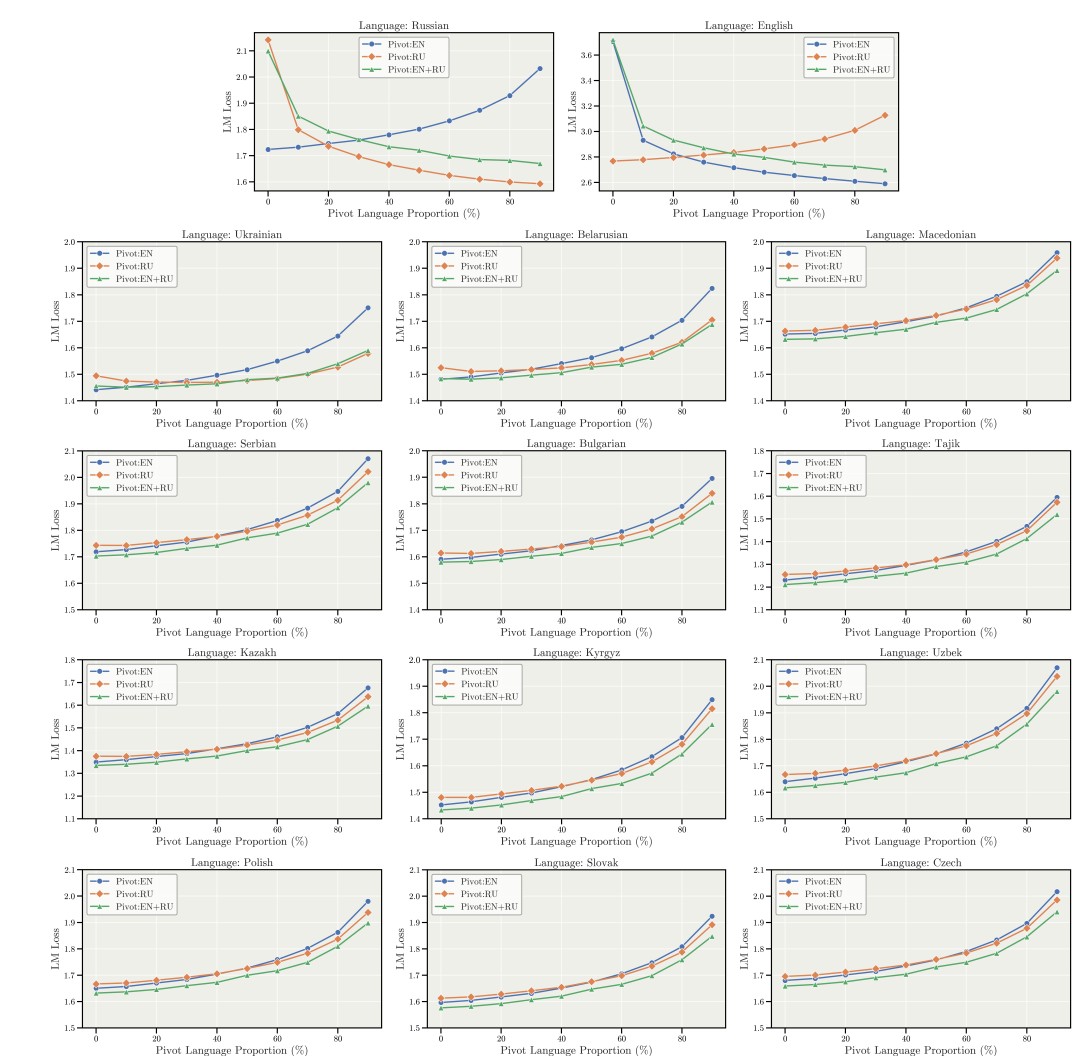

Figure 9: Validation LM loss for each language in the experiments described in Section 4, using English, Russian, or a combination of English and Russian as the pivot language in the training mix. The combination of English and Russian yields the best performance for most languages (model size: $1.1B$).

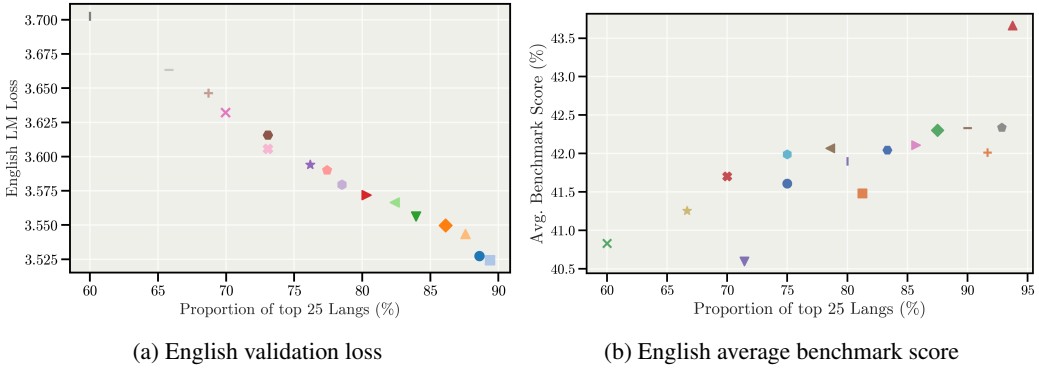

(a) English validation loss

(b) English average benchmark score

Figure 10: English (a) validation LM loss and (b) average benchmark score across different proportions of the top 25 languages (model size: $3B$). Increasing token allocation for tail languages reduces validation loss in English and improves English accuracy.

| Language Family | Pearson Correlation |
| --- | --- |
| Slavic | 0.853 |
| Germanic | 0.808 |
| Romance | 0.785 |
| Malayo-Sumbawan | 0.683 |
| Semitic | 0.521 |
| Creoles and Pidgins | –0.578 |
| Kuki-Chin | –0.759 |
| Bantu | –0.776 |
| Greater Central Philippine | –0.827 |
| Mixtec | –0.832 |
| Celebic | –0.872 |
| Cariban | –0.895 |
| Panoan | –0.897 |
| Oti-Volta | –0.899 |
| Western Mande | –0.915 |
| Zapotecan | –0.919 |
| Mayan | –0.925 |
| Brahmaputran | –0.939 |
| Northern Luzon | –0.945 |
| Chinantecan | –0.952 |
| Oceanic | –0.962 |
| Algonquian | –0.963 |
| Quechuan | –0.980 |
| Central Malayo-Polynesian | –0.985 |
| Maweti-Guarani | –0.985 |
| Tucanoan | –0.992 |

Table 5: Pearson correlation ($r$) between English validation loss and each language family, retaining only results with $p < 0.05$ and sorted in descending order of $r$.

| Script | # Languages |
|--------|-------------|
| Latn | 1639 |
| Cyrl | 56 |
| Arab | 30 |
| Deva | 29 |
| Ethi | 9 |
| Thai | 7 |
| Cans | 6 |
| Beng | 5 |
| Mymr | 5 |
| Hani | 5 |
| Telu | 3 |
| Hebr | 3 |
| Grek | 3 |
| Tibt | 3 |
| Tfng | 2 |
| Armn | 2 |
| Orya | 2 |
| Geor | 2 |
| Syrc | 2 |
| Laoo | 2 |
| Knda | 2 |

Table 6: Scripts and the number of languages each one supports. Sixteen other scripts are present in the FineWeb-2 dataset, each supporting one language.

| Language Family | # Languages |
|-----------------|-------------|
| Bantu | 73 |
| Oceanic | 67 |
| Mayan | 24 |
| Turkic | 22 |
| Indic | 22 |
| Creoles and Pidgins | 20 |
| Germanic | 17 |
| Tucanoan | 16 |
| Greater Central Philippine | 15 |
| Romance | 15 |
| Semitic | 14 |
| Mixtec | 13 |
| Slavic | 13 |
| Zapotecan | 12 |
| Central Malayo-Polynesian | 12 |
| Iranian | 11 |
| Oti-Volta | 11 |
| Malayo-Sumbawan | 11 |
| Kuki-Chin | 10 |
| Northern Luzon | 10 |
| Celebic | 9 |
| Quechuan | 9 |
| Maweti-Guarani | 9 |
| Dravidian | 8 |
| Brahmaputran | 8 |
| Panoan | 8 |
| Western Mande | 8 |
| Cariban | 8 |
| Algonquian | 8 |
| Chinantecan | 7 |

Table 7: Top language sub-families in FineWeb-2 and their number of associated languages. The classification is according to Dryer & Haspelmath (2013). Labels for 768 languages in FineWeb-2 were not available.

| Num PT Langs | Variant | Top 25 lang $B$ Tokens (Prop.) | Top 50 lang $B$ Tokens (Prop.) | Top 100 lang $B$ Tokens (Prop.) | Top 200 lang $B$ Tokens (Prop.) |
|---|---|---|---|---|---|
| 50 | Natural | 55.77 (0.56) | - | - | - |
| | Temp. | 40.15 (0.40) | - | - | - |
| | Natural – C | 60.08 (0.56) | - | - | - |
| | Temp. – C | 60.08 (0.40) | - | - | - |
| 100 | Natural | 55.07 (0.55) | 59.33 (0.59) | - | - |
| | Temp. | 30.51 (0.30) | 45.65 (0.46) | - | - |
| | Natural – C | 55.77 (0.55) | 60.08 (0.59) | - | - |
| | Temp. – C | 40.15 (0.30) | 60.08 (0.46) | - | - |
| 200 | Natural | 54.98 (0.55) | 59.23 (0.59) | 59.99 (0.60) | - |
| | Temp. | 25.28 (0.25) | 37.83 (0.38) | 49.79 (0.50) | - |
| | Natural – C | 55.07 (0.55) | 59.33 (0.59) | 60.08 (0.60) | - |
| | Temp. – C | 30.51 (0.25) | 45.65 (0.38) | 60.08 (0.50) | - |
| 400 | Natural | 54.97 (0.55) | 59.22 (0.59) | 59.98 (0.60) | 60.07 (0.60) |
| | Temp. | 22.07 (0.22) | 33.03 (0.33) | 43.47 (0.43) | 52.46 (0.52) |
| | Natural – C | 54.98 (0.55) | 59.23 (0.59) | 59.99 (0.60) | 60.08 (0.60) |
| | Temp. – C | 25.28 (0.22) | 37.83 (0.33) | 49.79 (0.43) | 60.08 (0.52) |

Table 8: Total number of tokens (in billions) and the corresponding proportions contributed by the top-25, 50, 100, and 200 languages. *Num PT Langs* refers to the total number of languages included during pretraining. *Natural* and *Temp.* represent natural sampling and temperature-based sampling, respectively, both conducted with a fixed token budget of 100B tokens. *Natural-C* and *Temp.-C* denote the same sampling strategies applied under the Controlled Growth setting, which uses a total of 90B tokens. English is excluded from the token counts and proportions.

| Num PT Langs | Variant | BB | M3E | MMMLU | PAWS-X | XCSQA | XCodah | XCopa | XSC | XWG |
|---|---|---|---|---|---|---|---|---|---|---|
| 25 | Natural | 38.22 | 38.70 | 30.75 | 49.60 | 35.70 | 51.67 | 66.80 | 75.10 | 65.60 |
| | Temp. | 33.67 | 33.20 | 27.52 | 45.70 | 31.20 | 37.33 | 62.00 | 63.60 | 54.90 |
| 50 | Natural | 37.44 | 38.60 | 30.91 | 49.00 | 33.00 | 51.67 | 67.00 | 73.60 | 66.50 |
| | Temp. | 32.33 | 33.50 | 27.51 | 55.90 | 31.60 | 38.00 | 61.80 | 65.30 | 55.80 |
| 100 | Natural | 37.67 | 37.60 | 30.72 | 50.40 | 34.10 | 51.67 | 69.40 | 74.80 | 65.10 |
| | Temp. | 32.22 | 33.90 | 26.75 | 55.20 | 30.90 | 38.67 | 61.20 | 63.20 | 54.60 |
| 200 | Natural | 37.44 | 37.20 | 30.54 | 54.00 | 31.80 | 52.33 | 66.20 | 75.00 | 65.40 |
| | Temp. | 31.67 | 32.80 | 27.07 | 43.70 | 28.80 | 37.33 | 62.00 | 62.40 | 55.40 |
| 400 | Natural | 37.33 | 38.60 | 30.61 | 55.20 | 35.30 | 53.33 | 68.60 | 73.80 | 64.20 |
| | Temp. | 31.22 | 29.70 | 26.78 | 55.40 | 24.70 | 34.67 | 57.20 | 62.30 | 56.50 |

Table 9: Benchmark scores (%) for English with varying number and sampling of 25–400 languages during pretraining. *Num PT Langs* refers to the total number of languages included during pretraining. *Natural* and *Temp.* represent natural sampling and temperature-based sampling, respectively, both conducted with a fixed token budget of 100B tokens. *BB*, *M3E*, *XSC*, and *XWG* denote the results for BeleBele, M3Exams, XStoryCloze, and XWinogrande respectively.

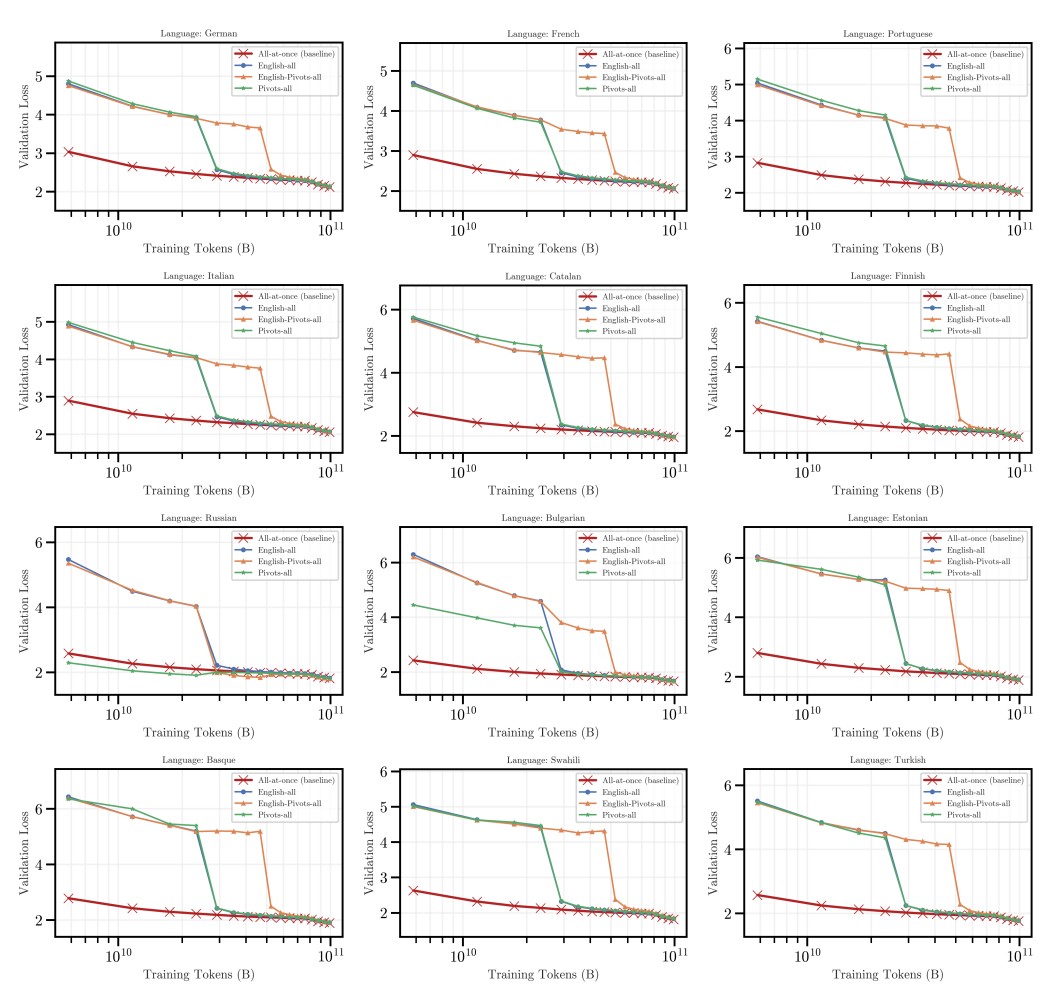

Figure 11: Validation LM loss (model size: 3B) for each language in the "Curriculum Learning" experiments described in Section 5 (Part 1).

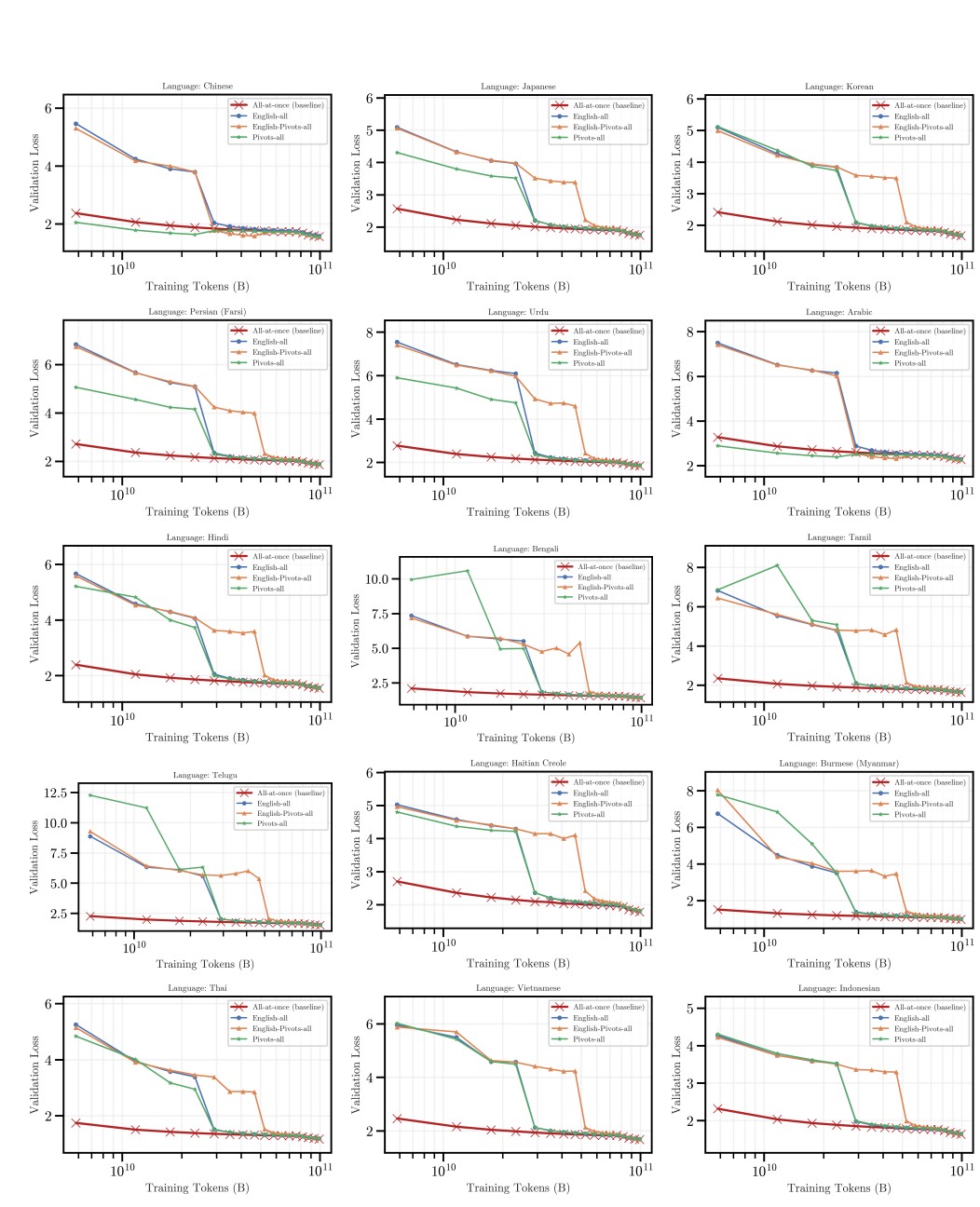

Figure 12: Validation LM loss (model size: 3B) for each language in the "Curriculum Learning" experiments described in Section 5 (Part 2).

| Language | en=00% | en=10% | en=20% | en=30% | en=40% | en=50% | en=60% | en=70% | en=80% | en=90% | en=100% |
|---|---|---|---|---|---|---|---|---|---|---|---|
| ar | 3810.73 (3.8%) | 3360.93 (3.4%) | 2987.49 (3.0%) | 2614.06 (2.6%) | 2286.44 (2.3%) | 1867.18 (1.9%) | 1434.00 (1.5%) | 1143.22 (1.1%) | 762.15 (0.8%) | 381.07 (0.4%) | 0.00 (0.0%) |
| bg | 2855.12 (2.9%) | 2609.73 (2.6%) | 2319.76 (2.3%) | 2029.79 (2.0%) | 1713.45 (1.7%) | 1449.85 (1.4%) | 1113.49 (1.2%) | 856.72 (0.9%) | 571.15 (0.6%) | 285.57 (0.3%) | 0.00 (0.0%) |
| bn | 2044.27 (2.0%) | 1850.78 (1.9%) | 1645.14 (1.6%) | 1439.50 (1.4%) | 1226.56 (1.2%) | 1028.21 (1.0%) | 789.67 (0.8%) | 613.28 (0.6%) | 408.85 (0.4%) | 204.43 (0.2%) | 0.00 (0.0%) |
| ca | 2434.91 (2.4%) | 2245.24 (2.2%) | 1995.77 (2.0%) | 1746.30 (1.7%) | 1460.95 (1.5%) | 1247.36 (1.2%) | 957.97 (1.0%) | 730.47 (0.7%) | 486.98 (0.5%) | 243.49 (0.2%) | 0.00 (0.0%) |
| de | 6587.51 (6.6%) | 6186.07 (6.2%) | 5499.35 (5.5%) | 4811.93 (4.8%) | 3952.51 (4.0%) | 3437.09 (3.4%) | 2639.69 (2.7%) | 1976.25 (2.0%) | 1317.50 (1.3%) | 658.75 (0.7%) | 0.00 (0.0%) |
| el | 3498.77 (3.5%) | 3132.17 (3.1%) | 2784.15 (2.8%) | 2436.13 (2.4%) | 2099.26 (2.1%) | 1740.89 (1.7%) | 1336.49 (1.4%) | 1049.63 (1.0%) | 699.75 (0.7%) | 349.88 (0.3%) | 0.00 (0.0%) |
| en | 0.00 (0.0%) | 10002.43 (10.0%) | 20004.86 (20.0%) | 30007.30 (30.0%) | 40009.73 (40.0%) | 50012.16 (50.0%) | 60014.59 (60.0%) | 70017.02 (70.0%) | 80019.46 (80.0%) | 90021.89 (90.0%) | 100024.32 (100.0%) |
| es | 7044.66 (7.0%) | 6275.04 (6.3%) | 5577.81 (5.6%) | 4880.58 (4.9%) | 4226.80 (4.2%) | 3486.13 (3.5%) | 2677.35 (2.8%) | 2113.40 (2.1%) | 1408.93 (1.4%) | 704.47 (0.7%) | 0.00 (0.0%) |
| et | 2009.65 (2.0%) | 1811.96 (1.8%) | 1610.63 (1.6%) | 1409.30 (1.4%) | 1205.79 (1.2%) | 1006.64 (1.0%) | 773.10 (0.8%) | 602.90 (0.6%) | 401.93 (0.4%) | 200.97 (0.2%) | 0.00 (0.0%) |
| eu | 1239.38 (1.2%) | 1163.61 (1.2%) | 1034.32 (1.0%) | 905.03 (0.9%) | 743.63 (0.7%) | 646.45 (0.6%) | 496.47 (0.5%) | 371.82 (0.4%) | 247.88 (0.2%) | 123.94 (0.1%) | 0.00 (0.0%) |
| fa | 3706.17 (3.7%) | 3380.02 (3.4%) | 3004.46 (3.0%) | 2628.91 (2.6%) | 2223.70 (2.2%) | 1877.79 (1.9%) | 1442.14 (1.5%) | 1111.85 (1.1%) | 741.23 (0.7%) | 370.62 (0.4%) | 0.00 (0.0%) |
| fi | 2968.54 (3.0%) | 2739.67 (2.7%) | 2435.26 (2.4%) | 2130.85 (2.1%) | 1781.12 (1.8%) | 1522.04 (1.5%) | 1168.93 (1.2%) | 890.56 (0.9%) | 593.71 (0.6%) | 296.85 (0.3%) | 0.00 (0.0%) |
| fr | 6415.58 (6.4%) | 5865.82 (5.9%) | 5214.06 (5.2%) | 4562.30 (4.6%) | 3849.35 (3.8%) | 3258.79 (3.3%) | 2502.75 (2.6%) | 1924.67 (1.9%) | 1283.12 (1.3%) | 641.56 (0.6%) | 0.00 (0.0%) |
| hi | 2932.04 (2.9%) | 2462.93 (2.5%) | 2189.27 (2.2%) | 1915.61 (1.9%) | 1759.23 (1.8%) | 1368.30 (1.4%) | 1050.85 (1.1%) | 879.61 (0.9%) | 586.41 (0.6%) | 293.20 (0.3%) | 0.00 (0.0%) |
| ht | 687.25 (0.7%) | 700.65 (0.7%) | 622.80 (0.6%) | 544.95 (0.5%) | 412.35 (0.4%) | 389.25 (0.4%) | 298.95 (0.3%) | 206.18 (0.2%) | 137.45 (0.1%) | 68.73 (0.1%) | 0.00 (0.0%) |
| id | 4037.86 (4.0%) | 3656.50 (3.7%) | 3250.27 (3.2%) | 2843.99 (2.8%) | 2422.72 (2.4%) | 2031.42 (2.0%) | 1560.13 (1.6%) | 1211.36 (1.2%) | 807.57 (0.8%) | 403.79 (0.4%) | 0.00 (0.0%) |
| it | 5229.67 (5.2%) | 4916.80 (4.9%) | 4370.49 (4.4%) | 3824.18 (3.8%) | 3137.80 (3.1%) | 2731.56 (2.7%) | 2097.84 (2.2%) | 1568.90 (1.6%) | 1045.93 (1.0%) | 522.97 (0.5%) | 0.00 (0.0%) |
| ja | 5249.15 (5.2%) | 3905.57 (3.9%) | 3471.62 (3.5%) | 3037.67 (3.0%) | 3149.49 (3.1%) | 2169.76 (2.2%) | 1666.38 (1.7%) | 1574.75 (1.6%) | 1049.83 (1.0%) | 524.92 (0.5%) | 0.00 (0.0%) |
| ko | 3004.03 (3.0%) | 2337.96 (2.3%) | 2078.18 (2.1%) | 1818.41 (1.8%) | 1802.42 (1.8%) | 1298.86 (1.3%) | 997.53 (1.0%) | 901.21 (0.9%) | 600.81 (0.6%) | 300.40 (0.3%) | 0.00 (0.0%) |
| my | 1084.07 (1.1%) | 943.16 (0.9%) | 838.36 (0.8%) | 733.57 (0.7%) | 650.44 (0.7%) | 523.98 (0.5%) | 402.41 (0.4%) | 325.22 (0.3%) | 216.81 (0.2%) | 108.41 (0.1%) | 0.00 (0.0%) |
| pt | 5067.45 (5.1%) | 4776.05 (4.8%) | 4245.38 (4.2%) | 3714.70 (3.7%) | 3040.47 (3.0%) | 2653.36 (2.7%) | 2037.78 (2.1%) | 1520.23 (1.5%) | 1013.49 (1.0%) | 506.74 (0.5%) | 0.00 (0.0%) |
| ru | 8194.02 (8.2%) | 7520.23 (7.5%) | 6684.65 (6.7%) | 5849.07 (5.8%) | 4916.41 (4.9%) | 4096.41 (4.1%) | 3208.63 (3.3%) | 2458.21 (2.5%) | 1638.80 (1.6%) | 819.40 (0.8%) | 0.00 (0.0%) |
| sw | 1119.24 (1.1%) | 1009.14 (1.0%) | 897.01 (0.9%) | 784.89 (0.8%) | 671.55 (0.7%) | 560.63 (0.6%) | 430.57 (0.4%) | 335.77 (0.3%) | 223.85 (0.2%) | 111.92 (0.1%) | 0.00 (0.0%) |
| ta | 1621.73 (1.6%) | 1475.10 (1.5%) | 1311.20 (1.3%) | 1147.30 (1.1%) | 973.04 (1.0%) | 819.50 (0.8%) | 629.37 (0.7%) | 486.52 (0.5%) | 324.35 (0.3%) | 162.17 (0.2%) | 0.00 (0.0%) |
| te | 1211.86 (1.2%) | 1066.46 (1.1%) | 947.97 (0.9%) | 829.47 (0.8%) | 727.12 (0.7%) | 592.48 (0.6%) | 455.02 (0.5%) | 363.56 (0.4%) | 242.37 (0.2%) | 121.19 (0.1%) | 0.00 (0.0%) |
| th | 2314.72 (2.3%) | 2292.68 (2.3%) | 2037.94 (2.0%) | 1783.19 (1.8%) | 1388.83 (1.4%) | 1273.71 (1.3%) | 978.21 (1.0%) | 694.42 (0.7%) | 462.94 (0.5%) | 231.47 (0.2%) | 0.00 (0.0%) |
| tr | 4072.98 (4.1%) | 3919.12 (3.9%) | 3483.66 (3.5%) | 3048.21 (3.0%) | 2443.79 (2.4%) | 2177.29 (2.2%) | 1672.16 (1.7%) | 1221.89 (1.2%) | 814.60 (0.8%) | 407.30 (0.4%) | 0.00 (0.0%) |
| ur | 1459.28 (1.5%) | 1225.81 (1.2%) | 1089.61 (1.1%) | 953.41 (1.0%) | 875.57 (0.9%) | 681.00 (0.7%) | 523.01 (0.5%) | 437.79 (0.4%) | 291.86 (0.3%) | 145.93 (0.1%) | 0.00 (0.0%) |
| vi | 4726.26 (4.7%) | 3793.06 (3.8%) | 3371.61 (3.4%) | 2950.16 (2.9%) | 2835.76 (2.8%) | 2107.26 (2.1%) | 1618.37 (1.7%) | 1417.88 (1.4%) | 945.25 (0.9%) | 472.63 (0.5%) | 0.00 (0.0%) |
| zh | 3396.76 (3.4%) | 3398.87 (3.4%) | 3021.22 (3.0%) | 2643.56 (2.6%) | 2038.06 (2.0%) | 1888.26 (1.9%) | 1450.18 (1.5%) | 1019.03 (1.0%) | 679.35 (0.7%) | 339.68 (0.3%) | 0.00 (0.0%) |

Table 10: Token counts (in millions) and their total proportions (%) for the *Fixed Total Budget* experiments described in Section 3. Total number of tokens is 100B.

| Language | en=20% | en=30% | en=40% | en=50% | en=60% |
|---|---|---|---|---|---|
| ar | 3345.99 (3.0%) | 3345.99 (2.6%) | 3316.12 (2.2%) | 3286.24 (1.9%) | 3345.99 (1.5%) |
| bg | 2598.14 (2.3%) | 2598.14 (2.0%) | 2574.94 (1.7%) | 2551.74 (1.4%) | 2598.14 (1.2%) |
| bn | 1842.56 (1.6%) | 1842.56 (1.4%) | 1826.10 (1.2%) | 1809.65 (1.0%) | 1842.56 (0.8%) |
| ca | 2235.26 (2.0%) | 2235.26 (1.7%) | 2215.31 (1.5%) | 2195.35 (1.2%) | 2235.26 (1.0%) |
| de | 6159.27 (5.5%) | 6159.27 (4.8%) | 6104.28 (4.1%) | 6049.28 (3.4%) | 6159.27 (2.7%) |
| el | 3118.25 (2.8%) | 3118.25 (2.4%) | 3090.41 (2.1%) | 3062.57 (1.7%) | 3118.25 (1.4%) |
| en | 22405.45 (20.0%) | 38409.34 (30.0%) | 59214.40 (40.0%) | 88021.40 (50.0%) | 134432.69 (60.0%) |
| es | 6247.15 (5.6%) | 6247.15 (4.9%) | 6191.37 (4.2%) | 6135.59 (3.5%) | 6247.15 (2.8%) |
| et | 1803.91 (1.6%) | 1803.91 (1.4%) | 1787.80 (1.2%) | 1771.69 (1.0%) | 1803.91 (0.8%) |
| eu | 1158.43 (1.0%) | 1158.43 (0.9%) | 1148.09 (0.8%) | 1137.75 (0.6%) | 1158.43 (0.5%) |
| fa | 3365.00 (3.0%) | 3365.00 (2.6%) | 3334.95 (2.3%) | 3304.91 (1.9%) | 3365.00 (1.5%) |
| fi | 2727.49 (2.4%) | 2727.49 (2.1%) | 2703.14 (1.8%) | 2678.79 (1.5%) | 2727.49 (1.2%) |
| fr | 5839.75 (5.2%) | 5839.75 (4.6%) | 5787.61 (3.9%) | 5735.47 (3.3%) | 5839.75 (2.6%) |
| hi | 2451.99 (2.2%) | 2451.99 (1.9%) | 2430.09 (1.6%) | 2408.20 (1.4%) | 2451.99 (1.1%) |
| ht | 697.54 (0.6%) | 697.54 (0.5%) | 691.31 (0.5%) | 685.08 (0.4%) | 697.54 (0.3%) |
| id | 3640.30 (3.2%) | 3640.30 (2.8%) | 3607.80 (2.4%) | 3575.30 (2.0%) | 3640.30 (1.6%) |
| it | 4894.95 (4.4%) | 4894.95 (3.8%) | 4851.25 (3.3%) | 4807.54 (2.7%) | 4894.95 (2.2%) |
| ja | 3888.21 (3.5%) | 3888.21 (3.0%) | 3853.50 (2.6%) | 3818.78 (2.2%) | 3888.21 (1.7%) |
| ko | 2327.57 (2.1%) | 2327.57 (1.8%) | 2306.78 (1.6%) | 2286.00 (1.3%) | 2327.57 (1.0%) |
| my | 938.97 (0.8%) | 938.97 (0.7%) | 930.58 (0.6%) | 922.20 (0.5%) | 938.97 (0.4%) |
| pt | 4754.82 (4.2%) | 4754.82 (3.7%) | 4712.37 (3.2%) | 4669.91 (2.7%) | 4754.82 (2.1%) |
| ru | 7486.80 (6.7%) | 7486.80 (5.8%) | 7419.96 (5.0%) | 7353.11 (4.2%) | 7486.80 (3.3%) |
| sw | 1004.66 (0.9%) | 1004.66 (0.8%) | 995.69 (0.7%) | 986.72 (0.6%) | 1004.66 (0.4%) |
| ta | 1468.54 (1.3%) | 1468.54 (1.1%) | 1455.43 (1.0%) | 1442.32 (0.8%) | 1468.54 (0.7%) |
| te | 1061.72 (0.9%) | 1061.72 (0.8%) | 1052.24 (0.7%) | 1042.76 (0.6%) | 1061.72 (0.5%) |
| th | 2282.49 (2.0%) | 2282.49 (1.8%) | 2262.11 (1.5%) | 2241.73 (1.3%) | 2282.49 (1.0%) |
| tr | 3901.70 (3.5%) | 3901.70 (3.0%) | 3866.87 (2.6%) | 3832.03 (2.2%) | 3901.70 (1.7%) |
| ur | 1220.36 (1.1%) | 1220.36 (1.0%) | 1209.46 (0.8%) | 1198.57 (0.7%) | 1220.36 (0.5%) |
| vi | 3776.21 (3.4%) | 3776.21 (2.9%) | 3742.49 (2.5%) | 3708.77 (2.1%) | 3776.21 (1.7%) |
| zh | 3383.76 (3.0%) | 3383.76 (2.6%) | 3353.55 (2.3%) | 3323.34 (1.9%) | 3383.76 (1.5%) |
| **Total** | 112027.20 (100.0%) | 128031.99 (100.0%) | 148037.75 (100.0%) | 176043.52 (100.0%) | 224054.06 (100.0%) |

Table 11: Token counts (in millions) and their total proportions (%) for the *Fixed Multilingual Budget* experiments described in Section 3.

| Language | en=00% | en=10% | en=20% | en=30% | en=40% | en=50% | en=60% | en=70% | en=80% | en=90% | en=100% |
|---|---|---|---|---|---|---|---|---|---|---|---|
| be | 3733.16 (3.7%) | 3359.85 (3.4%) | 2986.53 (3.0%) | 2613.21 (2.6%) | 2239.90 (2.2%) | 1866.58 (1.9%) | 1493.27 (1.5%) | 1119.95 (1.1%) | 746.63 (0.7%) | 373.32 (0.4%) | 0.00 (0.0%) |
| bg | 7720.59 (7.7%) | 6948.53 (6.9%) | 6176.47 (6.2%) | 5404.41 (5.4%) | 4632.36 (4.6%) | 3860.30 (3.9%) | 3088.24 (3.1%) | 2316.18 (2.3%) | 1544.12 (1.5%) | 772.06 (0.8%) | 0.00 (0.0%) |
| cs | 10619.66 (10.6%) | 9557.69 (9.6%) | 8495.73 (8.5%) | 7433.76 (7.4%) | 6371.80 (6.4%) | 5309.83 (5.3%) | 4247.86 (4.2%) | 3185.90 (3.2%) | 2123.93 (2.1%) | 1061.97 (1.1%) | 0.00 (0.0%) |
| en | 0.00 (0.0%) | 10002.43 (10.0%) | 20004.86 (20.0%) | 30007.30 (30.0%) | 40009.73 (40.0%) | 50012.16 (50.0%) | 60014.59 (60.0%) | 70017.02 (70.0%) | 80019.46 (80.0%) | 90021.89 (90.0%) | 100024.32 (100.0%) |
| kk | 4263.37 (4.3%) | 3837.04 (3.8%) | 3410.70 (3.4%) | 2984.36 (3.0%) | 2558.02 (2.6%) | 2131.69 (2.1%) | 1705.35 (1.7%) | 1279.01 (1.3%) | 852.67 (0.9%) | 426.34 (0.4%) | 0.00 (0.0%) |
| ky | 3025.91 (3.0%) | 2723.32 (2.7%) | 2420.73 (2.4%) | 2118.14 (2.1%) | 1815.55 (1.8%) | 1512.95 (1.5%) | 1210.36 (1.2%) | 907.77 (0.9%) | 605.18 (0.6%) | 302.59 (0.3%) | 0.00 (0.0%) |
| mk | 3615.86 (3.6%) | 3254.27 (3.3%) | 2892.69 (2.9%) | 2531.10 (2.5%) | 2169.51 (2.2%) | 1807.93 (1.8%) | 1446.34 (1.4%) | 1084.76 (1.1%) | 723.17 (0.7%) | 361.59 (0.4%) | 0.00 (0.0%) |
| mn | 4088.58 (4.1%) | 3679.72 (3.7%) | 3270.86 (3.3%) | 2862.00 (2.9%) | 2453.15 (2.5%) | 2044.29 (2.0%) | 1635.43 (1.6%) | 1226.57 (1.2%) | 817.72 (0.8%) | 408.86 (0.4%) | 0.00 (0.0%) |
| pl | 13226.50 (13.2%) | 11903.85 (11.9%) | 10581.20 (10.6%) | 9258.55 (9.3%) | 7935.90 (7.9%) | 6613.25 (6.6%) | 5290.60 (5.3%) | 3967.95 (4.0%) | 2645.30 (2.6%) | 1322.65 (1.3%) | 0.00 (0.0%) |
| ru | 22152.79 (22.1%) | 19937.51 (19.9%) | 17722.23 (17.7%) | 15506.95 (15.5%) | 13291.68 (13.3%) | 11076.40 (11.1%) | 8861.12 (8.9%) | 6645.84 (6.6%) | 4430.56 (4.4%) | 2215.28 (2.2%) | 0.00 (0.0%) |
| sk | 7265.10 (7.3%) | 6538.59 (6.5%) | 5812.08 (5.8%) | 5085.57 (5.1%) | 4359.06 (4.4%) | 3632.55 (3.6%) | 2906.04 (2.9%) | 2179.53 (2.2%) | 1453.02 (1.5%) | 726.51 (0.7%) | 0.00 (0.0%) |
| sr | 4707.78 (4.7%) | 4237.00 (4.2%) | 3766.22 (3.8%) | 3295.44 (3.3%) | 2824.67 (2.8%) | 2353.89 (2.4%) | 1883.11 (1.9%) | 1412.33 (1.4%) | 941.56 (0.9%) | 470.78 (0.5%) | 0.00 (0.0%) |
| tg | 3350.71 (3.3%) | 3015.64 (3.0%) | 2680.57 (2.7%) | 2345.50 (2.3%) | 2010.43 (2.0%) | 1675.36 (1.7%) | 1340.28 (1.3%) | 1005.21 (1.0%) | 670.14 (0.7%) | 335.07 (0.3%) | 0.00 (0.0%) |
| uk | 9323.48 (9.3%) | 8391.14 (8.4%) | 7458.79 (7.5%) | 6526.44 (6.5%) | 5594.09 (5.6%) | 4661.74 (4.7%) | 3729.39 (3.7%) | 2797.05 (2.8%) | 1864.70 (1.9%) | 932.35 (0.9%) | 0.00 (0.0%) |
| uz | 2930.83 (2.9%) | 2637.74 (2.6%) | 2344.66 (2.3%) | 2051.58 (2.1%) | 1758.50 (1.8%) | 1465.41 (1.5%) | 1172.33 (1.2%) | 879.25 (0.9%) | 586.17 (0.6%) | 293.08 (0.3%) | 0.00 (0.0%) |

Table 12: Token counts (in millions) and their total proportions (%) for the *english* as pivot runs described in Section 4. Total number of tokens is 100B.

| Language | ru=00% | ru=10% | ru=20% | ru=30% | ru=40% | ru=50% | ru=60% | ru=70% | ru=80% | ru=90% | ru=100% |
|---|---|---|---|---|---|---|---|---|---|---|---|
| be | 3359.26 (3.4%) | 3023.33 (3.0%) | 2687.41 (2.7%) | 2351.48 (2.4%) | 2015.56 (2.0%) | 1679.63 (1.7%) | 1343.70 (1.3%) | 1007.78 (1.0%) | 671.85 (0.7%) | 335.93 (0.3%) | 0.00 (0.0%) |
| bg | 6947.32 (6.9%) | 6252.59 (6.3%) | 5557.86 (5.6%) | 4863.12 (4.9%) | 4168.39 (4.2%) | 3473.66 (3.5%) | 2778.93 (2.8%) | 2084.20 (2.1%) | 1389.46 (1.4%) | 694.73 (0.7%) | 0.00 (0.0%) |
| cs | 9556.02 (9.6%) | 8600.42 (8.6%) | 7644.82 (7.6%) | 6689.22 (6.7%) | 5733.61 (5.7%) | 4778.01 (4.8%) | 3822.41 (3.8%) | 2866.81 (2.9%) | 1911.20 (1.9%) | 955.60 (1.0%) | 0.00 (0.0%) |
| en | 29952.18 (29.9%) | 26956.97 (27.0%) | 23961.75 (24.0%) | 20966.53 (21.0%) | 17971.31 (18.0%) | 14976.09 (15.0%) | 11980.87 (12.0%) | 8985.66 (9.0%) | 5990.44 (6.0%) | 2995.22 (3.0%) | 0.00 (0.0%) |
| kk | 3836.37 (3.8%) | 3452.73 (3.5%) | 3069.09 (3.1%) | 2685.46 (2.7%) | 2301.82 (2.3%) | 1918.18 (1.9%) | 1534.55 (1.5%) | 1150.91 (1.2%) | 767.27 (0.8%) | 383.64 (0.4%) | 0.00 (0.0%) |
| ky | 2722.84 (2.7%) | 2450.56 (2.4%) | 2178.27 (2.2%) | 1905.99 (1.9%) | 1633.71 (1.6%) | 1361.42 (1.4%) | 1089.14 (1.1%) | 816.85 (0.8%) | 544.57 (0.5%) | 272.28 (0.3%) | 0.00 (0.0%) |
| mk | 3253.70 (3.3%) | 2928.33 (2.9%) | 2602.96 (2.6%) | 2277.59 (2.3%) | 1952.22 (2.0%) | 1626.85 (1.6%) | 1301.48 (1.3%) | 976.11 (1.0%) | 650.74 (0.7%) | 325.37 (0.3%) | 0.00 (0.0%) |
| mn | 3679.08 (3.7%) | 3311.17 (3.3%) | 2943.26 (2.9%) | 2575.35 (2.6%) | 2207.45 (2.2%) | 1839.54 (1.8%) | 1471.63 (1.5%) | 1103.72 (1.1%) | 735.82 (0.7%) | 367.91 (0.4%) | 0.00 (0.0%) |
| pl | 11901.77 (11.9%) | 10711.59 (10.7%) | 9521.41 (9.5%) | 8331.24 (8.3%) | 7141.06 (7.1%) | 5950.88 (5.9%) | 4760.71 (4.8%) | 3570.53 (3.6%) | 2380.35 (2.4%) | 1190.18 (1.2%) | 0.00 (0.0%) |
| ru | 0.00 (0.0%) | 10002.43 (10.0%) | 20004.86 (20.0%) | 30007.30 (30.0%) | 40009.73 (40.0%) | 50012.16 (50.0%) | 60014.59 (60.0%) | 70017.02 (70.0%) | 80019.46 (80.0%) | 90021.89 (90.0%) | 100024.32 (100.0%) |
| sk | 6537.45 (6.5%) | 5883.70 (5.9%) | 5229.96 (5.2%) | 4576.21 (4.6%) | 3922.47 (3.9%) | 3268.72 (3.3%) | 2614.98 (2.6%) | 1961.23 (2.0%) | 1307.49 (1.3%) | 653.74 (0.7%) | 0.00 (0.0%) |
| sr | 4236.26 (4.2%) | 3812.63 (3.8%) | 3389.01 (3.4%) | 2965.38 (3.0%) | 2541.76 (2.5%) | 2118.13 (2.1%) | 1694.50 (1.7%) | 1270.88 (1.3%) | 847.25 (0.8%) | 423.63 (0.4%) | 0.00 (0.0%) |
| tg | 3015.11 (3.0%) | 2713.60 (2.7%) | 2412.09 (2.4%) | 2110.58 (2.1%) | 1809.07 (1.8%) | 1507.56 (1.5%) | 1206.05 (1.2%) | 904.53 (0.9%) | 603.02 (0.6%) | 301.51 (0.3%) | 0.00 (0.0%) |
| uk | 8389.67 (8.4%) | 7550.70 (7.5%) | 6711.74 (6.7%) | 5872.77 (5.9%) | 5033.80 (5.0%) | 4194.84 (4.2%) | 3355.87 (3.4%) | 2516.90 (2.5%) | 1677.93 (1.7%) | 838.97 (0.8%) | 0.00 (0.0%) |
| uz | 2637.28 (2.6%) | 2373.55 (2.4%) | 2109.83 (2.1%) | 1846.10 (1.8%) | 1582.37 (1.6%) | 1318.64 (1.3%) | 1054.91 (1.1%) | 791.18 (0.8%) | 527.46 (0.5%) | 263.73 (0.3%) | 0.00 (0.0%) |

Table 13: Token counts (in millions) and their total proportions (%) for the *russian* as pivot runs described in Section 4. Total number of tokens is 100B.

| Language | en=00%,ru=00% | en=05%,ru=05% | en=10%,ru=10% | en=15%,ru=15% | en=20%,ru=20% | en=25%,ru=25% | en=30%,ru=30% | en=35%,ru=35% | en=40%,ru=40% | en=45%,ru=45% | en=50%,ru=50% |
|---|---|---|---|---|---|---|---|---|---|---|---|
| be | 4795.17 (4.8%) | 4315.65 (4.3%) | 3836.14 (3.8%) | 3356.62 (3.4%) | 2877.10 (2.9%) | 2301.68 (2.4%) | 1918.07 (1.9%) | 1438.55 (1.4%) | 920.67 (1.0%) | 479.52 (0.5%) | 0.00 (0.0%) |
| bg | 9916.94 (9.9%) | 8925.24 (8.9%) | 7933.55 (7.9%) | 6941.86 (6.9%) | 5950.16 (5.9%) | 4760.13 (5.0%) | 3966.77 (4.0%) | 2975.08 (3.0%) | 1904.05 (2.0%) | 991.69 (1.0%) | 0.00 (0.0%) |
| cs | 13640.73 (13.6%) | 12276.65 (12.3%) | 10912.58 (10.9%) | 9548.51 (9.5%) | 8184.44 (8.2%) | 6547.55 (6.8%) | 5456.29 (5.5%) | 4092.22 (4.1%) | 2619.02 (2.7%) | 1364.07 (1.4%) | 0.00 (0.0%) |
| en | 0.00 (0.0%) | 5001.22 (5.0%) | 10002.43 (10.0%) | 15003.65 (15.0%) | 20004.86 (20.0%) | 24005.84 (25.0%) | 30007.30 (30.0%) | 35008.51 (35.0%) | 38409.34 (40.0%) | 45010.94 (45.0%) | 50012.16 (50.0%) |
| kk | 5476.21 (5.5%) | 4928.59 (4.9%) | 4380.97 (4.4%) | 3833.35 (3.8%) | 3285.73 (3.3%) | 2628.58 (2.7%) | 2190.48 (2.2%) | 1642.86 (1.6%) | 1051.43 (1.1%) | 547.62 (0.5%) | 0.00 (0.0%) |
| ky | 3886.72 (3.9%) | 3498.04 (3.5%) | 3109.37 (3.1%) | 2720.70 (2.7%) | 2332.03 (2.3%) | 1865.62 (1.9%) | 1554.69 (1.6%) | 1166.01 (1.2%) | 746.25 (0.8%) | 388.67 (0.4%) | 0.00 (0.0%) |
| mk | 4644.49 (4.6%) | 4180.04 (4.2%) | 3715.59 (3.7%) | 3251.14 (3.3%) | 2786.69 (2.8%) | 2229.36 (2.3%) | 1857.80 (1.9%) | 1393.35 (1.4%) | 891.74 (0.9%) | 464.45 (0.5%) | 0.00 (0.0%) |
| mn | 5251.69 (5.3%) | 4726.52 (4.7%) | 4201.35 (4.2%) | 3676.18 (3.7%) | 3151.01 (3.2%) | 2520.81 (2.6%) | 2100.68 (2.1%) | 1575.51 (1.6%) | 1008.32 (1.1%) | 525.17 (0.5%) | 0.00 (0.0%) |
| pl | 16989.15 (17.0%) | 15290.24 (15.3%) | 13591.32 (13.6%) | 11892.41 (11.9%) | 10193.49 (10.2%) | 8154.79 (8.5%) | 6795.66 (6.8%) | 5096.75 (5.1%) | 3261.92 (3.4%) | 1698.92 (1.7%) | 0.00 (0.0%) |
| ru | 0.00 (0.0%) | 5001.22 (5.0%) | 10002.43 (10.0%) | 15003.65 (15.0%) | 20004.86 (20.0%) | 24005.84 (25.0%) | 30007.30 (30.0%) | 35008.51 (35.0%) | 38409.34 (40.0%) | 45010.94 (45.0%) | 50012.16 (50.0%) |
| sk | 9331.86 (9.3%) | 8398.68 (8.4%) | 7465.49 (7.5%) | 6532.30 (6.5%) | 5599.12 (5.6%) | 4479.29 (4.7%) | 3732.75 (3.7%) | 2799.56 (2.8%) | 1791.72 (1.9%) | 933.19 (0.9%) | 0.00 (0.0%) |
| sr | 6047.04 (6.0%) | 5442.34 (5.4%) | 4837.63 (4.8%) | 4232.93 (4.2%) | 3628.22 (3.6%) | 2902.58 (3.0%) | 2418.82 (2.4%) | 1814.11 (1.8%) | 1161.03 (1.2%) | 604.70 (0.6%) | 0.00 (0.0%) |
| tg | 4303.92 (4.3%) | 3873.53 (3.9%) | 3443.13 (3.4%) | 3012.74 (3.0%) | 2582.35 (2.6%) | 2065.88 (2.2%) | 1721.57 (1.7%) | 1291.18 (1.3%) | 826.35 (0.9%) | 430.39 (0.4%) | 0.00 (0.0%) |
| uk | 11975.82 (12.0%) | 10778.24 (10.8%) | 9580.65 (9.6%) | 8383.07 (8.4%) | 7185.49 (7.2%) | 5748.39 (6.0%) | 4790.33 (4.8%) | 3592.75 (3.6%) | 2299.36 (2.4%) | 1197.58 (1.2%) | 0.00 (0.0%) |
| uz | 3764.58 (3.8%) | 3388.12 (3.4%) | 3011.67 (3.0%) | 2635.21 (2.6%) | 2258.75 (2.3%) | 1807.00 (1.9%) | 1505.83 (1.5%) | 1129.37 (1.1%) | 722.80 (0.8%) | 376.46 (0.4%) | 0.00 (0.0%) |

Table 14: Token counts (in millions) and their total proportions (%) for the *english and russian* as pivots runs described in Section 4. Total number of tokens is 100B.

| Language | Natural Distribution | | | | | Temperature Sampling | | | | |
|---|---|---|---|---|---|---|---|---|---|---|
| | 25 | 50 | 100 | 200 | 400 | 25 | 50 | 100 | 200 | 400 |
| arb | 1.95 | 1.95 | 1.95 | 1.94 | 1.94 | 1.83 | 1.85 | 1.87 | 1.90 | 1.93 |
| ces | 1.53 | 1.53 | 1.53 | 1.51 | 1.52 | 1.45 | 1.47 | 1.50 | 1.54 | 1.57 |
| cmn | 1.67 | 1.68 | 1.68 | 1.69 | 1.69 | 1.78 | 1.86 | 1.92 | 1.96 | 2.00 |
| dan | 1.49 | 1.48 | 1.49 | 1.49 | 1.49 | 1.39 | 1.42 | 1.44 | 1.47 | 1.49 |
| deu | 1.53 | 1.53 | 1.54 | 1.54 | 1.53 | 1.57 | 1.61 | 1.64 | 1.69 | 1.72 |
| ell | 1.20 | 1.20 | 1.20 | 1.20 | 1.20 | 1.12 | 1.15 | 1.17 | 1.19 | 1.22 |
| eng | 2.67 | 2.67 | 2.68 | 2.68 | 2.67 | 2.67 | 2.68 | 2.68 | 2.69 | 2.70 |
| fas | 1.61 | 1.62 | 1.62 | 1.62 | 1.62 | 1.52 | 1.56 | 1.59 | 1.61 | 1.63 |
| fra | 1.51 | 1.52 | 1.52 | 1.52 | 1.52 | 1.54 | 1.57 | 1.59 | 1.61 | 1.64 |
| hun | 1.70 | 1.70 | 1.71 | 1.70 | 1.70 | 1.58 | 1.61 | 1.65 | 1.69 | 1.72 |
| ind | 1.43 | 1.43 | 1.43 | 1.42 | 1.42 | 1.38 | 1.41 | 1.42 | 1.44 | 1.46 |
| ita | 1.52 | 1.52 | 1.53 | 1.52 | 1.52 | 1.52 | 1.56 | 1.58 | 1.61 | 1.64 |
| jpn | 1.34 | 1.35 | 1.35 | 1.35 | 1.35 | 1.38 | 1.42 | 1.46 | 1.49 | 1.52 |
| kor | 1.74 | 1.75 | 1.76 | 1.74 | 1.73 | 1.64 | 1.69 | 1.74 | 1.77 | 1.80 |
| nld | 1.35 | 1.36 | 1.36 | 1.36 | 1.36 | 1.33 | 1.36 | 1.39 | 1.40 | 1.43 |
| pol | 1.34 | 1.35 | 1.35 | 1.35 | 1.36 | 1.32 | 1.35 | 1.38 | 1.41 | 1.44 |
| por | 1.48 | 1.49 | 1.50 | 1.49 | 1.49 | 1.48 | 1.52 | 1.54 | 1.56 | 1.58 |
| ron | 1.47 | 1.48 | 1.48 | 1.48 | 1.48 | 1.39 | 1.43 | 1.46 | 1.49 | 1.51 |
| rus | 1.52 | 1.53 | 1.54 | 1.54 | 1.55 | 1.59 | 1.63 | 1.66 | 1.69 | 1.72 |
| spa | 1.46 | 1.46 | 1.47 | 1.47 | 1.46 | 1.48 | 1.51 | 1.53 | 1.55 | 1.57 |
| swe | 1.59 | 1.59 | 1.59 | 1.59 | 1.59 | 1.49 | 1.52 | 1.55 | 1.58 | 1.60 |
| tha | 1.27 | 1.28 | 1.28 | 1.28 | 1.28 | 1.16 | 1.20 | 1.23 | 1.25 | 1.27 |
| tur | 1.40 | 1.41 | 1.41 | 1.40 | 1.40 | 1.33 | 1.36 | 1.39 | 1.42 | 1.44 |
| ukr | 1.30 | 1.31 | 1.31 | 1.31 | 1.31 | 1.25 | 1.28 | 1.31 | 1.33 | 1.36 |
| vie | 1.55 | 1.56 | 1.56 | 1.56 | 1.56 | 1.45 | 1.49 | 1.52 | 1.55 | 1.57 |
| als | - | 1.50 | 1.50 | 1.50 | 1.50 | - | 1.29 | 1.32 | 1.34 | 1.36 |
| ary | - | 1.83 | 1.82 | 1.83 | 1.82 | - | 1.62 | 1.62 | 1.65 | 1.68 |
| azj | - | 1.47 | 1.47 | 1.48 | 1.48 | - | 1.07 | 1.10 | 1.12 | 1.15 |
| ben | - | 1.32 | 1.32 | 1.32 | 1.32 | - | 1.16 | 1.18 | 1.21 | 1.23 |
| bos | - | 1.44 | 1.45 | 1.45 | 1.45 | - | 1.31 | 1.35 | 1.38 | 1.40 |
| bul | - | 1.46 | 1.47 | 1.47 | 1.47 | - | 1.36 | 1.39 | 1.42 | 1.45 |
| cat | - | 1.36 | 1.36 | 1.36 | 1.36 | - | 1.28 | 1.31 | 1.33 | 1.35 |
| ekk | - | 1.83 | 1.83 | 1.83 | 1.83 | - | 1.56 | 1.61 | 1.64 | 1.68 |
| fin | - | 1.65 | 1.66 | 1.66 | 1.66 | - | 1.53 | 1.57 | 1.60 | 1.63 |
| heb | - | 1.70 | 1.71 | 1.71 | 1.71 | - | 1.52 | 1.55 | 1.58 | 1.62 |
| hin | - | 1.42 | 1.42 | 1.42 | 1.42 | - | 1.28 | 1.28 | 1.30 | 1.33 |
| hrv | - | 1.61 | 1.62 | 1.62 | 1.62 | - | 1.48 | 1.52 | 1.55 | 1.58 |
| kat | - | 1.40 | 1.41 | 1.41 | 1.40 | - | 1.11 | 1.14 | 1.16 | 1.19 |
| lit | - | 1.58 | 1.59 | 1.59 | 1.59 | - | 1.39 | 1.42 | 1.46 | 1.49 |
| lvs | - | 1.59 | 1.59 | 1.59 | 1.59 | - | 1.35 | 1.38 | 1.42 | 1.45 |
| mar | - | 1.52 | 1.53 | 1.52 | 1.53 | - | 1.23 | 1.25 | 1.27 | 1.30 |
| mkd | - | 1.43 | 1.44 | 1.44 | 1.44 | - | 1.20 | 1.23 | 1.26 | 1.28 |
| nob | - | 1.92 | 1.92 | 1.92 | 1.92 | - | 1.82 | 1.85 | 1.88 | 1.91 |
| npi | - | 1.32 | 1.33 | 1.32 | 1.32 | - | 1.08 | 1.10 | 1.12 | 1.14 |
| slk | - | 1.46 | 1.46 | 1.46 | 1.46 | - | 1.36 | 1.40 | 1.43 | 1.46 |
| slv | - | 1.65 | 1.66 | 1.66 | 1.66 | - | 1.46 | 1.50 | 1.54 | 1.57 |
| srp | - | 1.79 | 1.79 | 1.79 | 1.79 | - | 1.49 | 1.53 | 1.56 | 1.59 |
| tam | - | 1.52 | 1.52 | 1.52 | 1.52 | - | 1.28 | 1.29 | 1.32 | 1.34 |
| urd | - | 1.77 | 1.78 | 1.78 | 1.78 | - | 1.46 | 1.48 | 1.50 | 1.53 |
| zsm | - | 1.62 | 1.62 | 1.63 | 1.62 | - | 1.46 | 1.48 | 1.51 | 1.53 |

Table 15: Comparison of the final validation loss for 50 languages in the *curse of multilinguality* experiments. The results are presented for a fixed total data budget under two conditions: natural distribution and temperature sampling.

