# OpenReview forum: "Revisiting Multilingual Data Mixtures in Language Model Pretraining"
_ICLR.cc/2026/Conference — Submitted to ICLR 2026_

### Official Review · Reviewer_CfNz · 2025-10-30

**Soundness:** 2
**Presentation:** 3
**Contribution:** 2
**Rating:** 4
**Confidence:** 4

**Summary:**

The paper investigates how multilingual data mixtures affect LLM pretraining, focusing on the "curse of multilinguality." The authors train 1B–3B parameter models with up to 400 languages using the mC4 and FineWeb2 datasets. They test four key assumptions:

1. English hurts multilinguality: Increasing English data does not necessarily degrade non-English performance.

2. Family-specific pivots help: Using English as a pivot language performs as well as or better than family-specific pivots.

3. Curriculum learning helps: The order of language introduction does not significantly reduce interference.

4. Curse of multilinguality: Adding more languages does not inherently hurt performance if data is balanced.

Their conclusion: Multilingual data, when balanced appropriately, can enhance language model
capabilities without compromising performance, even in low-resource settings.

**Strengths:**

- Comprehensive experimental design: Systematic testing of four major hypotheses at multiple scales (1B and 3B models).

- Relevant and timely topic: Multilingual data efficiency and scaling are crucial for LLM research.

- Clear presentation: Good structure, consistent visualizations, and explicit takeaways per assumption.

- Empirical insights: Provides evidence that challenges widespread assumptions about multilinguality trade-offs.

**Weaknesses:**

- Questionable assumptions / missing motivation: The origins of “Assumptions 1–4” are not sufficiently grounded in prior literature. For example, a citation of the works that assumed or discussed Assumption 1 is not listed.

- Tokenizer bias as a confounder: The use of the fixed Mistral-Nemo-Base-2407 tokenizer, which primarily includes English tokens, introduces a major confounding factor. Since all languages are tokenized with this fixed vocabulary, tokenization efficiency and representation differ across languages. This could significantly skew the results toward English, especially in Assumptions 1 and 2.

- Missing related work: Recent peer-reviewed work on the same topic, but focused on continual multilingual pretraining (https://arxiv.org/pdf/2504.04152, COLM 2025), is omitted.

- Citations around L304 on the number of languages these models are trained on are needed. The citations are sometimes misformatted (e.g., L572 “Josh Gpt-4 Team”?).

- You can also define the curse of multilinguality for a language or a group of languages: some may benefit when other languages are added, while others degrade. Showing only the overall trend that “the curse” is absent in your setting may not be very informative. You might not have enough training tokens to reach the point where performance degrades and adding more languages is not helpful. It would be useful to highlight some specific languages that are hurt and others that benefit as well (see https://aclanthology.org/2023.acl-long.61.pdf, Table 9).


- Limited findings: Typically, models are trained on many languages. If practitioners are already including multiple languages and with increasing the proportion of English data (aligned with the findings suggestions): how can these results inform or influence current pretraining practices for such models?

**Questions:**

- It is hard to believe that adding more multilingual data in Assumption 1 (under a fixed total budget) does not significantly improve downstream multilingual performance. If that is indeed the case, how are models expected to improve their performance in multilingual settings, and why does this occur?


- Could retraining with a multilingual tokenizer, or using a separate tokenizer for each experiment, alter the conclusions? How much do tokenizer biases affect results, especially for non-Latin scripts?

- You have two experiments on the imbalance between languages (natural and temperature) in assumption 4, but I still could not figure out whether the total number of tokens for each experiment is 100B. Does this mean that if there are 25 languages, the total is 100B, and if there are 400 languages, it is still 100B? We need experiments for both cases: one where the total size is fixed and one where the total size grows with the number of languages.

- If the order does not matter, what type of curriculum learning helps?

---

> ### Author Response · Authors · 2025-11-24
>
> We thank the reviewer for their positive feedback, particularly for recognizing the comprehensive nature of our experimental design and the clarity of our presentation. We are also pleased they found our research questions to address a relevant and timely topic. Below, we address their concerns, re-formatting them for easier future reference.
>
> ## [W1] Missing citations for motivating each assumption:
> Research on data sampling strategies demonstrates that training on the 'natural' distribution of web data—which is predominantly English—leads to substantial degradation in multilingual zero-shot performance. This necessitates the use of artificial 'temperature sampling' to down-weight English and up-weight lower-resource languages to prevent parameter starvation. We dive more deeply into this tradeoff in our work. We added missing citations to the corresponding section [2,3,4].
>
> **Assumption 2:** Previous research established that cross-lingual transfer is significantly influenced by the linguistic distance between languages, often resulting in asymmetric performance [5, 6, 7,8]. Specifically, performance is generally higher between languages from the same language family. Our contribution investigates this phenomenon specifically within the pre-training setting. We test whether using a pivot language from the same family as the target languages yields better pre-training outcomes, and show that English as a pivot language is just as effective.
>
> **Assumption 3:** Prior research has explored using curriculum learning (a 'general-to-specific' data scheduling approach) to improve pre-training (Llama3, Databricks DBRX, Apertus, EuroLMs). In multilingual training, it has been claimed that the order in which languages are introduced can impact model performance and potentially reduce competition among them [9,10,11]. We investigate this hypothesis for pre-training and provide evidence that, contrary to prior claims, the order of languages does not significantly affect performance during the initial pre-training stage.
>
> **Assumption 4:**
> The concept known as the "curse of multilinguality" has been extensively explored in prior research [12, 13, 14,15]. This work distinguishes itself by conducting a comprehensive and integrated analysis of the phenomenon across its various facets.
>
>
>
> ## [W2] Tokenizer effect
> We acknowledge that choice of tokenizer is a confounding factor in multilingual modeling. We selected the Mistral-Nemo-Base-2407 tokenizer because it is a state-of-the-art tokenizer designed specifically for multilingual pretraining (trained using more than 100 languages), covering a much wider range of scripts than other publicly available tokenizers as shown in [1]. While training language-specific tokenizers for every ablation would theoretically isolate tokenization effects, it is computationally prohibitive given the number of experiments. Furthermore, using a single, strong multilingual tokenizer reflects the realistic standard practice for training Foundation Models, making our results relevant to current practices. We also note that the reviewer’s concern that tokenizers might be skewed toward English is a reflection of the reality of modern tokenizers.
>
> ## [W3] Suggested related work:
> Thanks for introducing this paper. We’ve included it in the related work section (lines 464-465).
>
> ## [W4] Citation problem around L304
> Thanks for pointing it out, we fixed the issue.
>
> ## [W5] Curse of multilinguality for a language or a group of languages:
> We agree with the reviewer's point that the "curse of multilinguality" can be also understood when examining specific language groups rather than just the overall average. While the main body of our work focuses on the average loss across all languages, we also conducted a separate, in-depth analysis for each individual language. Crucially, we did not observe any discernible pattern indicating that certain language groups or individual languages behave differently (i.e., benefiting or suffering more) when additional languages are introduced. For full transparency, the validation loss for all 50 languages across various experiments is presented in Table 15 of Appendix G.

---

> > ### Author Response · Authors · 2025-11-24
> >
> > ## [W6] Practitioners are already using multilingual data
> > While recent models have increasingly incorporated multilingual data, most models are still conservative in language coverage. Our work shifts the focus to the scientific principles underlying this progress as it's still unknown what the performance characteristics are after the inclusion of these languages. Beyond encouraging multilingual coverage through our experiments showing that English performance can be maintained in multilingual training, and that the ``curse of multilinguality’’ has limitations, we deliver three core messages for expanding multilingual coverage: (1)  there is no need for curriculum learning to boost multilingual performance; (2) there is no need to rely on linguistic genealogy for achieving superior cross-lingual results; and (3), a focus on data quality across all languages is required for a better multilingual performance.
> >
> > ## [Q1] More multilingual tokens doesn’t increase multilingual performance
> > We already observe a trend in Figure 2a, that more multilingual tokens does increase multilingual performance.To see a more significant performance boost across all 29 languages, we would need considerably more data for each language. However, these experiments focused specifically on showing that adding more English data does not negatively impact the other languages. While we recognize that more tokens are required to maximize overall multilingual performance, this was not feasible within our current computational budget.
> >
> > ## [Q2] Can better tokenizers change the results?
> > A “better” tokenizer across languages would represent more complex tokens with more mutual information between consecutive tokens. We'd expect overall performance to increase as the model learns more information on average from each token (assuming an equivalent token training budget). However, we don't foresee how this would change our overall results. We note that the Mistral tokenizer is already one of the fairest tokenizers across languages. Increasing the vocabulary size considerably would also scale up the amount of compute needed for each training step, which might be impractical for large training runs.
> > Optimizing tokenizers, whether multilingual or monolingual, can certainly improve performance across all languages, especially those using non-Latin scripts. A better tokenizer, for instance, might lower overall loss and mitigate negative interference by achieving better alignment across languages with similar scripts. However, crucial to our argument is that the choice of a highly multilingual tokenizer does not fundamentally alter the nature of language interaction and their respective effects on one another, meaning the overall conclusions of this work remain unchanged.
> >
> > ## [Q3] Total number of tokens
> > We explore both settings suggested by the reviewer. In Figure 5a and 5c, the total training budget is fixed at 100B tokens across all language counts (25, 50, 100, 200, and 400). In Figures 5b and 5d, we disentangle the effect of adding new languages from the reduction of per-language data. Here, the data volume for the original set of languages remains constant across consecutive runs. Consequently, the total data budget increases when expanding the language set (e.g., moving from 25 to 50 languages).
> >
> > ## [Q4] Effect of ordering data:
> > Previous work showed that curriculum learning defined as data scheduling or annealing where we give ‘general-to-specific' data to the model actually can help the pre-training. For example, Llama 3 explicitly used an annealing phase where they shifted the data mix toward high-quality code and reasoning in the final stages. A similar approach has been used in EuroLM and Apertus models. Databricks DBRX also noted using curriculum learning to change data mixtures during training to boost quality.

---

> > > ### Author Response · Authors · 2025-11-24
> > >
> > > **References:**
> > >
> > > [1] Apertus: Democratizing Open and Compliant LLMs for Global Language Environments (arxiv 2025)
> > >
> > > [2] UniMax: Fairer and more Effective Language Sampling for Large-Scale Multilingual Pretraining (ICLR 2023)
> > >
> > > [3] mT5: A massively multilingual pre-trained text-to-text transformer (NAACL 2021)
> > >
> > > [4] ByT5: Towards a Token-Free Future with Pre-trained Byte-to-Byte Models (TACL 2022)
> > >
> > > [5] Scaling laws for multilingual language models (ACL 2025)
> > >
> > > [6] What drives performance in multilingual language models? (2024)
> > >
> > > [7] Languages you know influence those you learn: Impact of language characteristics on multi-lingual text-to-text transfer (PMLR 2023)
> > >
> > > [8] Cross-Lingual Pitfalls: Automatic Probing Cross-Lingual Weakness of Multilingual Large Language Models (ACL 2025)
> > >
> > > [9] Order matters in the presence of dataset imbalance for multilingual learning (NeurIPS 2023)
> > >
> > > [10] Optimizing the Training Schedule of Multilingual NMT using Reinforcement Learning (mtsummit 2025)
> > >
> > > [11] Does the Order Matter? Curriculum Learning over Languages (LREC-COLING 2024)
> > >
> > > [12] Unsupervised Cross-lingual Representation Learning at Scale (ACL 2020)
> > >
> > > [13] Lifting the Curse of Multilinguality by Pre-training Modular Transformers (ACL 2022)
> > >
> > > [14] Breaking the Curse of Multilinguality with Cross-lingual Expert Language Models (EMNL 2024)
> > >
> > > [15] When is multilinguality a curse? (EMNLP 2024)

---

> > > > ### Comment · Reviewer_CfNz · 2025-11-26
> > > > **response to response**
> > > >
> > > > Thank you for your detailed response to the weaknesses and questions! While I appreciate this additional information I would like to keep my overall rating.

---

### Official Review · Reviewer_Vkmh · 2025-10-30

**Soundness:** 3
**Presentation:** 3
**Contribution:** 3
**Rating:** 4
**Confidence:** 4

**Summary:**

This paper investigates some questions related to multilingual data mixtures for LLM pre-training by training 1–3B parameter LLMs on multilingual corpora covering 25–400 languages.

The examined assumptions and corresponding findings are:
1\. More English data comes at the cost of performance in other languages. -> Increasing the amount of English data does not necessarily degrade multilingual performance.

2\. Languages within the same family offer the strongest boost to multilingual generalization. -> English serves as a broadly effective pivotal language but in low-resource settings, typologically similar pivots can be important.

3\. Curriculum-based language introduction mitigates negative interference. -> Curriculum learning does not have an observed practical impact.

4\. Adding more languages to a pretraining mixture reduces performance. -> Performance reduction is due to finite capacity of models rather than the addition of languages.

**Strengths:**

1. The study investigates a number of different research questions that can be taken into consideration when pre-training large multilingual models.

2. The paper studies models at two different parameter sizes, which improves the generality of the findings.

3. The study investigates scaling with different numbers of languages, including a larger number of languages.

4. Ablation studies are generally well-thought out, such as comparing fixed total and fixed multilingual budget or studying pivot languages from different language families.

**Weaknesses:**

As a whole, I have concerns regarding the relevance of the studied assumptions and as a result the novelty of the corresponding insights.

Re #1: recent closed-source and open-source models have much improved English and multilingual performance, which indicates that more English data does not come at the cost of performance in other languages (up to a %), contrary to the assumption stated in the paper.

Re #3: curriculum learning has not been used in the pre-training of a state-of-the-art LLM as far as I’m aware, so finding that curriculum learning does not help is not surprising IMO. Studies that demonstrate the circumstances under which curriculum learning can be effective (such as this one: [https://arxiv.org/abs/2506.11300](https://arxiv.org/abs/2506.11300)) seem more surprising in that regard.

Re #4: The finding that the curse of multilinguality relates to model capacity rather than simply adding more languages has already been stated in the original XLM-R paper introducing the term. The XLM-R authors write: “Model capacity […] is constrained due to practical considerations […]. For a fixed size model, the per-language capacity decreases as we increase the number of languages.” This observation is also what motivated and led to the success of adapters for multilingual model adaptation. So I’m unsure of the novelty of the finding given the consensus in prior work.

I think the study would be stronger if you highlighted which findings are novel more clearly and explored the actually novel or practically relevant ones more in-depth. Right now, the findings that are already established or well-known take attention away from the other parts.

For the study of assumption #1, I am missing a comparison to a setting where the model is only trained on English data so that English performance is maximized. This reflects the most common setting in practice where it is key to compete on English performance and multilingual performance is increased as long as it does not decrease English performance. Extending the graphs in Figures 1 and 2 to English Data Proportion 100% would provide this comparison.

A confounding factor in the study of multilingual data mixtures is the tokenizer. A tokenizer with low compression rates for some languages leads to the model seeing less data (as more tokens are necessary to represent each data point) for those languages. So in addition to looking at the relative token budget across languages, it would be good to control for the tokenizer’s compression rate in some experiments. Alternatively, doing similar experiments with another tokenizer (with a markedly different compression distribution) would also shed additional light on this. Here are a few studies on the effects of tokenization on language models: [https://arxiv.org/abs/2012.15613](https://arxiv.org/abs/2012.15613), [https://aclanthology.org/2023.emnlp-main.614/](https://aclanthology.org/2023.emnlp-main.614/)

As has been shown in the past, the behavior of models can change dramatically with larger parameter sizes. The examined assumptions such as the curse of multilinguality or the trade-off between English and multilingual performance are also closely tied to model capacity. Running experiments on a model in the 20–40B range would provide evidence that the provided findings generalize to sizes of models that are more commonly used in practice these days.

The current experimental setting restricts training to a single epoch as far as I’m aware. This doesn’t capture the impact of repeated data in pre-training, which is relevant for under-represented languages where repeating data may be necessary to see a sufficient amounts of tokens. Prior work ([https://arxiv.org/abs/2305.16264](https://arxiv.org/abs/2305.16264)) has shown that in data-constrained settings, repeating data up to 4 times performs similarly to training on equivalent amounts of unique data. It would be useful to understand to what the number of training epochs impacts the paper’s findings.

**Questions:**

Line 368: What does it mean for performance to remain stable here? Is this based on a weighted or uniform average of performance across languages?

---

> ### Author Response · Authors · 2025-11-24
>
> We thank the reviewer for their positive assessment, particularly for acknowledging the well-thought nature of our experiments and the usefulness of our research questions for multilingual training. We have synthesized the remaining comments and questions and provide our detailed responses below, re-formatting them for easier future reference.
>
> ## [W1] Many models are not trained with multilingual data, so English data not hurting multilingual performance is common knowledge:
> While recent models incorporate multilingual data, the scientific literature still lacks rigorous ablation studies that precisely quantify the trade-off of when English helps versus when it hurts. For instance, earlier work [2, 3, 4] highlighted negative interference as a major issue. We refine this understanding by demonstrating that interference is primarily a function of the available training token budget, not simply the inclusion of other languages. Moreover, current multilingual training is often treated as a limited, auxiliary add-on, typically including data for only the top 20–30 languages so the model can generate tokens in them. In contrast, our work pushes beyond this limited scope. Our experiments decisively show that high multilingual performance can be achieved simultaneously with high English performance. We thus encourage practitioners to embrace comprehensive multilingual training without the fear of sacrificing English fluency.
>
> ## [W3] curriculum learning has not been used in the pre-training of LLMs:
> A broad form of curriculum learning, e.g., data scheduling or annealing, is becoming standard in SOTA models. For example, Llama 3 explicitly used an annealing phase where they shifted the data mix toward high-quality code and reasoning in the final stages. A similar approach has been used in EuroLM and Apertus models. Databricks DBRX also noted using curriculum learning to change data mixtures during training to boost quality. In our study, we transferred the assumption of this success to multilingual generalization based on a curriculum defined in terms of the order of languages. Our negative result that elaborate CL strategies yield no benefit over joint training at scale is a valuable contribution that saves future researchers from pursuing dead ends. While this can potentially be viewed as a negative result, we think it is important to report it.
>
> ## [W4] Curse of capacity is already mentioned in XLM-R paper:
> XLM-R stated that capacity was the constraint, but their experimental setup conflated adding languages with reducing per-language capacity. They did not run a "Controlled Growth" experiment (like our Figure 5b) where the per-language budget is held fixed while the language count grows. By doing so, we mathematically isolate the interference effect and prove that language count per se is not the curse. This empirical distinction is a novel methodological contribution over XLM-R. In other words, the observation is not necessarily new, but in this work, in contrast to the previous studies, we investigate different aspects of this phenomenon as a whole (Added this in a new Discussion section).
>
> ## [W5] 100% English baseline for pivot studies:
> We updated figure 1 and 2 (fixed multilingual budget) by adding the 100% English baseline.
>
> ## [W6] Effect of tokenizer:
> We acknowledge that choice of tokenizer is a confounding factor in multilingual modeling. We selected the Mistral-Nemo-Base-2407 tokenizer because it is a state-of-the-art tokenizer designed specifically for multilingual pretraining, covering a wide range of scripts and languages (more than 100), and representing them more fairly than other publicly available tokenizers [1]. While training language-specific tokenizers for every ablation would theoretically isolate tokenization effects, it is computationally prohibitive given the number of experiments. Furthermore, using a single, strong multilingual tokenizer reflects the realistic standard practice for training Foundation Models, making our results relevant to current practices.
>
>
> ## [W7] # Scaling up to  20–40B models:
> We agree with the reviewer that scaling our experiments to larger models would provide valuable evidence regarding the generalizability of our findings. However, our current experimental rigor is already pushing the limits of the resource budget we have for this project. Executing the suggested experiment (which would involve models 10x larger than our largest current runs) is unfortunately beyond the computational and resource budget currently available.

---

> > ### Author Response · Authors · 2025-11-24
> >
> > ## [W8] The effect of training on repeated tokens:
> > In our natural-distribution setting, we did not oversample any language; the amount of data used for each language was limited by its actual availability. Conversely, in the temperature sampling setting, some languages were inherently oversampled due to the scarcity of their original data (the majority less than 4 times). However, given the computation-heavy nature of the experiments, we couldn’t manage to run more experiments to control for the number of repeated tokens for each language. We expect that repeating the training tokens after a threshold can hurt the performance of the model, regardless of which language the data comes from, and we agree that knowing this limit could be valuable for the research community.
> >
> >
> > ## [Q1] Line 368: What does it mean for performance to remain stable?
> > It refers to the weighted average LM loss (or benchmark score) across the non-English languages. "Stable" means the variance is negligible or within the margin of error, indicating no statistically significant degradation.
> >
> >
> > **References:**
> >
> > [1] Apertus: Democratizing Open and Compliant LLMs for Global Language Environments (arxiv 2025)
> >
> > [2] Interference matrix: Quantifying cross-lingual interference in transformer encoders (arxiv 2025)
> >
> > [3] On negative interference in multilingual models: Findings and a meta-learning treatment (EMNLP 2020)
> >
> > [4] When Is Multilinguality a Curse? Language Modeling for 250 High- and Low-Resource Languages (EMNLP 2024)

---

### Official Review · Reviewer_GiHe · 2025-11-02

**Soundness:** 3
**Presentation:** 3
**Contribution:** 3
**Rating:** 8
**Confidence:** 4

**Summary:**

The paper primarily studies the impact of multilingual data mixtures during pretraining under various conditions on downstream performance. It challenges several previously held beliefs about multilingual pretraining, showing that (i) when both pivot languages and less represented languages are present in sufficient quantities (even if their ratios are unequal), the model can perform well on both types of languages, and (ii) the so-called “curse of multilinguality” arises mainly due to limited model capacity and the artificial amplification of low-resource data points.

**Strengths:**

1. Tackles an important yet relatively understudied topic of multilingual fairness in large language models.
2. The paper is well written. The narrative is clear, and the experiments are comprehensive and carefully designed.

Overall, this is a strong paper that will be of interest to the community. I recommend acceptance, assuming the minor concerns outlined below are addressed during the rebuttal.

**Weaknesses:**

1. The pivot language experiments are limited to the Cyrillic and Slavic languages. It would strengthen the paper to include other language families to confirm the generality of the results, especially given that one of the paper’s main claimed strengths is its broad coverage of languages.
2. Table 1 could be made stronger by studying varying percentages of English instead of keeping it fixed at 40%. If the claim is that beyond a certain English data threshold, the number of additional languages has minimal effect, then I would expect to see comparisons across different proportions of English data.
3. I am interested to know what Figure 4c would look like on the non-pivot (multilingual) tasks. I think this would be particularly interesting to see as opposed to just the pivot languages because the model is explicitly trained on those anyway.

**Questions:**

1. In Figures 1 and 2, why is the validation loss for the multilingual model lower than that for the English-only model, even when English constitutes the majority of the data?
2. At the point where the two curves intersect (if they ever) in the unconstrained data setting (Figure 1b), is the multilingual loss still stable?

---

> ### Author Response · Authors · 2025-11-24
>
> We thank the reviewer for their insightful feedback and for considering this study important and finding the paper well-written and clear.
> We address the reviewer’s concerns here:
> ## [W1] More languages for pivot experiments:
> We acknowledge that a broader range of language families would enhance the generalizability of our pivot experiments. While constrained by computational limitations, our language selection was a strategic choice as the chosen languages represent a linguistically significant unit and are a classification supported by prior work [1,2,3]. The mix includes a combination of a high-resource language (Russian) and several other written languages that meet a critical data volume threshold necessary to draw statistically sound and meaningful conclusions in our work.
> ## [W2] Curse of multilinguality experiments for different proportions of English:
> We agree that including more experiments where English proportion changes as well would be valuable. However, given our limited computation budget and the need to  control the interaction between English and other languages, we decided to keep the English data fixed to have fewer changing parameters in this budget. Regarding the claim referenced by the reviewer, we note that most LLMs train on more than 40% English, so we assume our results would remain relevant in the practical contexts where more data was allocated to English.
> ## [W3] Per-language loss in curriculum learning experiments:
> Figure 4b currently shows the weighted average loss for non-pivot languages. It shows that while curricula change the path, they converge to the same destination (final loss). We included per-language validation loss in Appendix F in figures 12 and 13 and we observe the same pattern for every language in the mix.
>
>
> ## [Q1] Why is the validation loss higher for the English-only model ?
> The observed gap in the range of validation loss between English and multilingual data is likely due to differences in tokenization. Tokenizers are usually more efficient at tokenizing English, and thus use fewer, more semantically-loaded tokens (e.g., full words) that are harder for the model to predict than the smaller, more predictable subword pieces seen in the tokenized form of other lower-resource languages. This explains why most multilingual models exhibit higher perplexity (and thus loss) for English compared to other individual languages in the set. While the Mistral-Nemo-Base-2407 tokenizer is a fairer tokenizer than most other alternatives, it retains a more efficient tokenization in English compared to lower-resource languages.
>
>
> ## [Q2] Would the multilingual loss remain stable in unconstrained data settings if the English loss were to intersect it ?
> We expect, in this asymptotic case, that if  the model has enough capacity to jointly encode both all the English data and the limited multilingual data, we expect the loss to be stable, if it is uniformly distributed throughout the English data.
>
> **References:**
>
> [1] Cross-lingual Named Entity Corpus for Slavic Languages (LREC-COLING 2024)
>
> [2] Typological Features for Multilingual Delexicalised Dependency Parsing (NAACL 2019)
>
> [3] NLP for preserving Torlak, a vulnerable low-resource Slavic language (COLING 2025)

---

### Official Review · Reviewer_nKHM · 2025-11-04

**Soundness:** 4
**Presentation:** 4
**Contribution:** 2
**Rating:** 6
**Confidence:** 3

**Summary:**

This paper trains a number of multilingual language models to investigate several previously documented assumptions about how multilingual data mixtures affect a model’s quality. In particular, the paper:
* Trains multilingual models with a fixed amount of non-English data, but while varying the amount of English. It finds that the amount of English does not affect performance in non-English languages.
* Trains multilingual models with English vs. other languages used as “pivot” languages. They find that language family is only important for transfer in lower-resource scenarios.
* Trains multilingual models with multiple curricula (e.g., starting with English, then introducing other pivot languages, then lower-resource languages). This does not seem to significantly affect the performance in lower-resource languages.
* Trains multilingual models with varying numbers of languages. As long as the amount of data per language is kept fixed, performance is not harmed by increasing the number of languages.

**Strengths:**

The paper provides a “short survey” of several prior results on how to design multilingual data mixtures for training multilingual language models. The paper then provides evidence contrary to important beliefs propagated by these prior papers.

The paper is very well written and organised, with clear assumptions and takeaways from each experiment.

**Weaknesses:**

While this paper examines several prior assumptions about multilingual data mixtures, each assumption is not necessarily comprehensively examined. Further, the paper does not provide enough experiments to show *why* its results differ from prior works.

**Questions:**

> Selecting a high-resource pivot language from within a specific family (e.g., Russian for Slavic languages) does not consistently enhance performance across languages in that family

As far as I know, Slavic is not a language family per se, but a branch in the Indo-European language family. It is thus still in the same language family (i.e., Indo-European) as English (which is from the West Germanic branch of the Indo-European family).


> Figures.

The grey background present in many figures makes them harder to read when the paper is printed in black and white. I’d suggest removing it.

>  English Hurts Multilinguality

Related to this result. Wendler et al. (2024) show that large models can leverage one (pivot) language’s circuits when processing other languages. And Schäfer et al. (2024) showed that performance might transfer from a pivot to non-pivot languages when using imbalance language mixtures, but not when training models in a balanced setting.


* Wendler et al. 2024. [Do Llamas Work in English? On the Latent Language of Multilingual Transformers](https://aclanthology.org/2024.acl-long.820/). In: ACL.
* Schäfer et al. 2024. [The Role of Language Imbalance in Cross-lingual Generalisation: Insights from Cloned Language Experiments.](https://arxiv.org/abs/2404.07982). In: arXiv.

---

> ### Author Response · Authors · 2025-11-24
>
> We thank the reviewer for their valuable feedback and for finding the paper well-written, organized and with clear research questions and takeaways. We address the reviewer’s concerns here, re-formatting them for easier future reference:
>
> ## [W1] Why our results are different from some previous work:
> We believe our results differ primarily because of the scale and the experimental control (isolating token count vs. language count). For example, regarding the "Curse of Multilinguality," prior works often observed degradation because adding languages implicitly reduced the tokens available for each language under a fixed budget. By explicitly testing a "Fixed Multilingual Budget" (Fig 1b) and "Controlled Growth" (Fig 5b), we show that the "curse" disappears when capacity/tokens are not diluted.
>
>
> ## [Q1] Language Family Taxonomy for Slavic and Cyrillic languages:
> The reviewer suggests that Slavic languages, being part of the Indo-European (IE) family, are inherently similar to English. We acknowledge their shared IE ancestry, but the Slavic languages (part of the Balto-Slavic branch of IE) exhibits a grammatical structure that is significantly more divergent from English than many other IE sub-families [1]. Furthermore, the use of the Cyrillic script in a number of these languages introduces a fundamental difference in orthography, further separating them from Latin-script English. As a result, while Slavic Languages share a common ancestor to English as part of the IE family, we believe this experimental setup is appropriate for  showing the limits of linguistic genealogy as a driver of multilingual generalization. That being said, we are open to suggestions on how to better frame this setup if the reviewer thinks “language family” is too ambiguous a term to describe the commonalities among Slavic languages as a contrast to English.
> ## [Q2] Background of the plots:
> Thank you for pointing this out. We already removed the grey background in some plots to ensure they are printer-friendly. We’ll do the same for the remaining ones as well.
> ## [Q3] Suggested citations: Thanks for the suggested citations. We found them very relevant and added them to the related work section (Lines 489-492).
> [1] The Slavonic Languages (Bernard Comrie, Greville G. Corbett, 2003)

---

> ### Comment · Reviewer_nKHM · 2025-11-26
>
> I thank the authors for their response.
>
> > **[W1] Why our results are different from some previous work:**
> >
> > For example, regarding the "Curse of Multilinguality," prior works often observed degradation because adding languages implicitly reduced the tokens available for each language under a fixed budget. By explicitly testing a "Fixed Multilingual Budget" (Fig 1b) and "Controlled Growth" (Fig 5b), we show that the "curse" disappears when capacity/tokens are not diluted.
>
> This is helpful. Could you list the reasons why your results for the other three assumptions differ from prior work as well? I think making this extra clear on the paper would be helpful for readers. If you can list convincing reasons for all four assumptions, I would increase my score.
>
> > **[Q1] Language Family Taxonomy for Slavic and Cyrillic languages:**
> >
> > The reviewer suggests that Slavic languages, being part of the Indo-European (IE) family, are inherently similar to English.
> >
> > That being said, we are open to suggestions on how to better frame this setup if the reviewer thinks “language family” is too ambiguous a term to describe the commonalities among Slavic languages as a contrast to English.
>
> Sorry if my review wasn't clear. I didn't mean to suggest that Slavic languages are inherently similar to English or to criticise that part of your experimental design. My comment was regarding the accuracy of the text. Slavic languages are in the same language family as English, so that paragraph is technically incorrect and should be corrected. I think it should not be hard to rephrase that paragraph to avoid referring to language families. In fact, I think it would be important to rephrase every sentence which uses the term "family" imprecisely in the paper.

---

> > ### Author Response · Authors · 2025-12-03
> >
> > We thank the reviewer for their response and appreciate their engagement.
> >
> > ## [W1] Why our results are different from some previous work:
> > **Assumption 1 & 4:** The reasons our experimental results differ from prior work for these assumptions are essentially the same. Earlier studies did not conduct a controlled analysis that disentangles the effect of multilingual data from the effect of English-only data. In contrast, by using two experimental setups (Fixed Total Budget and Fixed Multilingual Budget (Figures 1 and 2)) we explicitly control for the trade-off between multilingual data and English data, allowing us to isolate their respective contributions.
> >
> > **Assumption 2:** Prior work on linguistic similarity has largely examined cross-lingual transfer during fine-tuning or in zero-shot prompting settings [2,3,4]. Our work instead investigates this phenomenon at the pretraining stage and shows that at this stage, linguistic similarity does not necessarily produce the expected transfer benefits. The only study that touches on this question during pretraining (the multilingual scaling laws work) analyzes 85M-parameter models while simultaneously varying the ratios of several languages within the same family [4]. Their setup includes limited coverage (4-5 languages per family, 23 languages in total) and their analysis focuses on correlations between average losses across families. By contrast, our experiments examine transfer from a single pivot language to others, which lets us directly control for within-family effects and isolate cross-language transfer in a more targeted way.
> >
> > **Assumption 3:** Previous curriculum-learning research has primarily focused on English [8,9,10], and within multilingual settings has mostly investigated curriculum effects during fine-tuning or SFT [5,6]. The only work that studies language ordering during pretraining [7] considers a highly imbalanced scenario in machine translation and multilingual training: one stage of pretraining on high-resource languages followed by fine-tuning on a mixture of high- and low-resource languages (across five languages). In contrast, our work extends the number of languages and examines curriculum learning exclusively during the pretraining phase, isolating it from confounding effects introduced during fine-tuning.
> >
> > ## [Q1]  Language Family Taxonomy for Slavic and Cyrillic languages:
> > Thanks for clarification on the language family term. Based on your input, we've revised the text to ensure precision. We replaced all imprecise uses with the more accurate terms: "language branch/group" or "linguistically/typologically similar languages."
> >
> > **References:**
> >
> > [2] What Drives Performance in Multilingual Language Models? (VarDial 2024)
> >
> > [3] Languages You Know Influence Those You Learn: Impact of Language Characteristics on Multi-Lingual Text-to-Text Transfer (PMLR 2023)
> >
> > [4] Scaling Laws for Multilingual Language Models (ACL 2025)
> >
> > [5] Optimizing the Training Schedule of Multilingual NMT using Reinforcement Learning (mtsummit 2025)
> >
> > [6] Does the Order Matter? Curriculum Learning over Languages (LREC-COLING 2024)
> >
> > [7] Order Matters in the Presence of Dataset Imbalance for Multilingual Learning (NeurIPS 2023)
> >
> > [8] Beyond Random Sampling: Efficient Language Model Pretraining via Curriculum Learning (arxiv 2025)
> >
> > [9] Preference Curriculum: LLMs Should Always Be Pretrained on Their Preferred Data (ACL 2025)
> >
> > [10] Dataset Decomposition: Faster LLM Training with Variable Sequence Length Curriculum (NeurIPS 2024)

---

### Author Response · Authors · 2025-11-24

We thank all reviewers for their constructive feedback and for recognizing the importance of our study on multilingual data mixtures. We are encouraged that reviewers found the paper "well written and organised", "comprehensive and carefully designed", and appreciated the "broad coverage of languages" and systematic ablations.
We address two common themes raised by multiple reviewers below, followed by specific responses to each reviewer.
The Choice of Tokenizer (Reviewers Vkmh, CfNz):

1. We acknowledge that choice of tokenizer is a confounding factor in multilingual modeling. We selected the Mistral-Nemo-Base-2407 tokenizer because it is a state-of-the-art tokenizer designed specifically for multilingual pretraining, covering a wide range of scripts and languages (over 100), and representing them more fairly than other publicly available tokenizers [1].
While training language-specific tokenizers for every ablation would theoretically isolate tokenization effects, it is computationally prohibitive given the number of experiments. Furthermore, using a single, strong multilingual tokenizer reflects the realistic standard practice for training Foundation Models, making our results relevant to current practices.

2. Novelty of Findings (Reviewers nKHM, Vkmh)
Some reviewers noted that certain findings (e.g., that English helps, or that the "curse" is capacity-related) align with intuition or hypotheses in prior work (like XLM-R). We respectfully emphasize that while these papers may have hypothesized that the curse of multilinguality was a capacity issue, they did not empirically disentangle language count from token budget per language at the billion-parameter scale. Our "Controlled Growth" experiments (Figure 5b/d) explicitly isolate these variables (at much greater scale), providing the empirical support that was previously missing. Similarly, while SOTA models use English, the limit of how much English can be added before hurting low-resource languages has rarely been mapped out systematically (Figure 1). We believe experimentally validating these assumptions is a crucial contribution to the science and development of multilingual LLMs.

[1] Apertus: Democratizing Open and Compliant LLMs for Global Language Environments (arxiv 2025)

---

### Author Response · Authors · 2025-12-03
**General Response**

We would first like to express our sincere gratitude to all the reviewers and the Area Chair for their time, insightful feedback, and constructive engagement with our paper. Their valuable comments were instrumental in strengthening the clarity and rigor of our work.

# Paper Summary:

Our paper challenged several prevailing assumptions about pretraining data mixtures for multilingual LLMs (1.1B and 3B models). First, we show that combining English and multilingual data does not degrade performance if languages have sufficient training tokens. Furthermore, using English as a pivot language yields benefits across language family branches, and contrary to expectations, selecting a pivot from within a specific language family branch does not consistently improve performance for that branch. Crucially, we observe no significant "curse of multilinguality" at this scale, suggesting that appropriately balanced multilingual data can enhance LLM capabilities without compromising performance, even in low-resource settings.

# Rebuttal Summary:
We thank the reviewers for their constructive feedback and suggestions. We appreciate their recognition of our work’s strengths in:

- **Topic Importance:** The research addresses an important, timely, and understudied topic (Reviewers GiHe, CfNz).
- **Evaluation & Experiments:** The experiments are viewed as comprehensive and carefully designed (CfNz, GiHe), including well-thought out ablations (Vkmh)
- **Presentation:** Well written and organised, with clear assumptions and takeaways from each experiment (nKHM, GiHe, ​​CfNz).

Major reviewer concerns focused on how and why the results of this work are different from previous work and to what extent our findings are grounded in literature. There were also questions about the effect of tokenizer choice on the findings of the paper, and suggestions on how to use the term “language family” in our experiments and some citation suggestions.

**Rebuttal Outcome:** During the initial discussion period, we addressed these concerns. Reviewer nKHM was convinced by our initial response was willing to increase their score from 6 → 8 given answering another question. However, we couldn’t continue the discussion due to the recent changes in the rebuttal process. Reviewer CfNz thanked us for our response and decided to maintain their score without further discussion or question. The other two reviewers had not yet engaged in the discussion by the time the changes reverted.

Below, we summarize reviewer feedback and our responses for each reviewer in more detail.

We addressed a question raised by multiple reviewers.
## The reviewers (Vkmh, CfNz) ask for the effect of tokenizer on our results:
The Mistral-Nemo-Base-2407 tokenizer was chosen for this work as it is designed for multilingual pretraining, covering over 100 languages more fairly than other available options. While we did not vary the effect of the tokenizer to contain experimental costs, we did not expect that changing the tokenizer to other reasonable options would fundamentally change the core conclusions of this work. A better multilingual tokenizer would use more complex tokens with higher mutual information, potentially improving model performance as more information is learned per token, but would also significantly increase the vocabulary size, making large training runs impractical due to higher computational costs. We did not expect either of these dynamics to dramatically change our findings.

---

> ### Author Response · Authors · 2025-12-03
> **Reviewer nKHM Summary:**
>
> # [W1]: The reviewer asked why our results were different from some prior work:
>
> We showed our results differ primarily because of the scale and the experimental control (isolating token count vs. language count). For example, regarding the "Curse of Multilinguality," prior works often observed degradation because adding languages implicitly reduced the tokens available for each language under a fixed budget. By explicitly testing a "Fixed Multilingual Budget" (Fig 1b) and "Controlled Growth" (Fig 5b), we show that the "curse" disappears when capacity/tokens are not diluted. We provided detailed explanation about each assumption and included extra information to the draft.
>
> https://openreview.net/forum?id=IKJyRyHpHV&noteId=KK2IMW0gvx
>
>
> [Q1] The reviewer asked to revise the use of “language family” in our text
> Based on their input, we've revised the text to ensure precision. We replaced all imprecise uses with the more accurate terms: "language branch/group" or "linguistically/typologically similar languages."
>
> https://openreview.net/forum?id=IKJyRyHpHV&noteId=Bld4UITjr6

---

> ### Author Response · Authors · 2025-12-03
> **Reviewer GiHe Summary**
>
> # [W1] The reviewer questioned why our  pivot experiments were limited to Slavic and Cyrillic languages:
>
> While constrained by computational limitations, our language selection was a strategic choice as the chosen languages represent a linguistically significant unit and are a classification supported by prior work [1,2,3].
>
> https://openreview.net/forum?id=IKJyRyHpHV&noteId=r0LxDJsxwn
>
> # [W2] There reviewer thinks the curse of multilinguality experiments for different proportions of English would make the results stronger:
> Given our limited computation budget and the need to control the interaction between English and other languages, we decided to keep the English data fixed to have fewer changing parameters in this budget. Regarding the claim referenced by the reviewer, we note that most LLMs train on more than 40% English, so we assume our results would remain relevant in the practical contexts where more data was allocated to English.
>
> Other minor concerns were addressed in the rebuttal around experimental observations (W3 & Q1).
>
> https://openreview.net/forum?id=IKJyRyHpHV&noteId=r0LxDJsxwn
>
>
> **References:**
>
> [1] Cross-lingual Named Entity Corpus for Slavic Languages (LREC-COLING 2024)
>
> [2] Typological Features for Multilingual Delexicalised Dependency Parsing (NAACL 2019)
>
> [3] NLP for preserving Torlak, a vulnerable low-resource Slavic language (COLING 2025)

---

> ### Author Response · Authors · 2025-12-03
> **Reviewer Vkmh Summary**
>
> # [W1] The reviewer points out that many models are now trained with multilingual data, so English data not hurting multilingual performance is common knowledge:
> While earlier models showed that multilingual data could cause negative interference, our research refines this by demonstrating that interference is primarily limited by the training token budget. Contrary to current limited practices, our experiments demonstrate that comprehensive multilingual training can achieve high performance in both English and other languages simultaneously. We encourage full adoption of multilingual training without concern for sacrificing English fluency.
>
> https://openreview.net/forum?id=IKJyRyHpHV&noteId=TAVMRAAONd
>
> # [W3] The reviewer challenges whether curriculum learning is really used in pre-training multilingual LLMs:
> A broad form of curriculum learning, e.g., data scheduling or annealing, is becoming standard in SOTA models (e.g., Llama 3, Databricks DBRX,  EuroLM and Apertus). In our study, we transferred the assumption of this success to multilingual generalization based on a curriculum defined in terms of the order of languages.
>
> https://openreview.net/forum?id=IKJyRyHpHV&noteId=TAVMRAAONd
>
> # [W4] The reviewer suggests that the curse of capacity is already mentioned in prior work (e.g., XLM-R):
> The constraint identified in XLM-R was a result of their experimental setup, which conflated increasing the number of languages with decreasing the capacity per language. We address this by introducing a "Controlled Growth" experiment (unlike XLM-R), which fixed the per-language capacity while growing the language count. This methodology allowed us to isolate and prove that the interference effect, not the sheer number of languages, is the performance bottleneck, marking a novel methodological distinction from previous studies.
>
> https://openreview.net/forum?id=IKJyRyHpHV&noteId=TAVMRAAONd
>
>
> # [W7] The reviewer mentions that scaling up to 20–40B models can change the results:
> Our current experimental rigor is already pushing the limits of the resource budget we have for this project. Executing the suggested experiment (which would involve models 10x larger than our largest current runs) is unfortunately beyond the computational and resource budget currently available.
>
> https://openreview.net/forum?id=IKJyRyHpHV&noteId=TAVMRAAONd
>
>
> # [W8] The reviewer suggested that training on repeated tokens could change our results:
> We noted that, in the temperature sampling setting, some languages were repeated due to the scarcity of their original data (the majority less than 4 times). However, given the computation-heavy nature of the experiments, we couldn’t manage to run more experiments to control for the number of repeated tokens for each language. We expect that repeating the training tokens after a threshold can hurt the performance of the model, regardless of which language the data comes from, and we agree that knowing this limit could be valuable for the research community.
>
> https://openreview.net/forum?id=IKJyRyHpHV&noteId=VegBISYBUX

---

> ### Author Response · Authors · 2025-12-03
> **Reviewer CfNz Summary**
>
> # [W5] The reviewer suggests that the curse of multilinguality can be also understood for individual languages, rather than all languages:
> The main body of our work focuses on the average loss across all languages, but we also conducted a separate, in-depth analysis for each individual language. We did not observe any discernible pattern indicating that certain language groups or individual languages behave differently (i.e., benefiting or suffering more) when additional languages are introduced. **For full transparency, the validation loss for all 50 languages across various experiments is presented in Table 15 of Appendix G.**
>
> https://openreview.net/forum?id=IKJyRyHpHV&noteId=D5VyoUcDJr
>
> # [W6] The reviewer questioned our assumptions relative to the context that practitioners are already using multilingual data in training LLMs
> We responded that most multilingual models still exhibit limited language coverage, despite increasing the inclusion of diverse data. Our research explores the scientific basis for this progress, specifically investigating the performance characteristics after adding new languages. We demonstrate that English performance is maintained during multilingual training, challenging the "curse of multilinguality," and offer three key findings for expanding coverage: (1) Curriculum learning is unnecessary; (2) Linguistic genealogical similarity is not essential for cross-lingual generalization; and (3) Data quality across all languages is the critical factor for superior multilingual performance.
>
> https://openreview.net/forum?id=IKJyRyHpHV&noteId=7BVRiXCCm2
>
>
> # [Q4] The reviewer asked if the order of the data doesn’t matter, what type of CL helps
> Previous work showed that curriculum learning defined as data scheduling or annealing where we give ‘general-to-specific' data to the model actually can help the pre-training. For example, Llama 3 explicitly used an annealing phase where they shifted the data mix toward high-quality code and reasoning in the final stages. A similar approach has been used in EuroLM and Apertus models. Databricks DBRX also noted using curriculum learning to change data mixtures during training to boost quality.
>
> Other minor concerns were addressed in the rebuttal around experimental observations (Q1 & Q3)
> https://openreview.net/forum?id=IKJyRyHpHV&noteId=7BVRiXCCm2

---

> ### Author Response · Authors · 2025-12-03
> **Thank You ACs for your Time and Consideration**
>
> We sincerely thank the Area Chairs for their time and effort in evaluating our submission. We appreciate the opportunity to address reviewer concerns and improve our manuscript through this process.

---

### Meta-Review · Area_Chair_QiQS · 2025-12-17

**Summary:**

This paper investigates multilingual data mixture strategies for LLM pretraining by training 1B and 3B parameter models on diverse multilingual corpora spanning 25-400 languages, challenging several prevailing assumptions about the "curse of multilinguality." The primary strengths include comprehensive and carefully designed experiments with systematic ablation studies comparing "Fixed Total Budget" vs. "Fixed Multilingual Budget" setups that explicitly isolate the trade-off between English and multilingual data, and "Controlled Growth" experiments that disentangle language count from per-language token budget—a novel methodological distinction over prior work like XLM-R. The paper is well-written and organized with clear assumptions and takeaways from each experiment, making it accessible and valuable for practitioners.

However, significant concerns remain about novelty and practical relevance. Multiple reviewers (nKHM, Vkmh, CfNz) noted that several findings appear to validate intuitions or hypotheses already present in prior work—for example, that recent closed/open-source models demonstrate English and multilingual performance can coexist, that XLM-R already identified capacity constraints as the curse mechanism, and that curriculum learning hasn't been standard in SOTA LLM pretraining (though authors note data scheduling/annealing is increasingly common in Llama 3, DBRX, EuroLM, Apertus). Reviewer Vkmh maintained a score of 4 despite comprehensive rebuttals, expressing concerns that "findings that are already established or well-known take attention away from other parts" and suggesting the paper would be stronger by highlighting truly novel findings more clearly. Reviewer nKHM's willingness to increase score to 8 was contingent on providing reasons why all four assumptions differ from prior work, which authors addressed but discussion was cut short by rebuttal process changes. Computational limitations prevented several requested extensions including experiments on 20-40B models (10x larger than current 3B), controlling for repeated token effects in temperature sampling settings (some languages repeated <4 times due to scarcity), and training separate tokenizers for ablations.

Given the above existing concerns, I recommend rejection, as the paper would benefit from further refinement to clearly delineate novel contributions from established knowledge and to enhance practical relevance through larger-scale experiments. The authors are encouraged to address reviewer feedback and consider resubmission in future venues.

**Reviewer Concerns:**

Two of the reviewers still remain unconvinced after the rebuttal.

**Reviewer Scores:**

Mixed, but two of them still remain negative after rebuttal.

---

### Decision · Program_Chairs · 2026-01-26

Reject